# 6D Object Pose Tracking in Internet Videos for Robotic Manipulation

**Georgy Ponimatkin**[1,2*]  **Martin Cífka**[1,2*]  **Tomáš Souček**[1]  **Médéric Fourmy**[1]
**Yann Labbé**[3]  **Vladimír Petrík**[1]  **Josef Sivic**[1]

[1]Czech Institute of Informatics, Robotics and Cybernetics, Czech Technical University in Prague
[2] Faculty of Electrical Engineering, Czech Technical University in Prague
[3] H Company

## Abstract

We seek to extract a temporally consistent 6D pose trajectory of a manipulated object from an Internet instructional video. This is a challenging set-up for current 6D pose estimation methods due to uncontrolled capturing conditions, subtle but dynamic object motions, and the fact that the exact mesh of the manipulated object is not known. To address these challenges, we present the following contributions. First, we develop a new method that estimates the 6D pose of any object in the input image without prior knowledge of the object itself. The method proceeds by (i) retrieving a CAD model similar to the depicted object from a large-scale model database, (ii) 6D aligning the retrieved CAD model with the input image, and (iii) grounding the absolute scale of the object with respect to the scene. Second, we extract smooth 6D object trajectories from Internet videos by carefully tracking the detected objects across video frames. The extracted object trajectories are then retargeted via trajectory optimization into the configuration space of a robotic manipulator. Third, we thoroughly evaluate and ablate our 6D pose estimation method on YCB-V and HOPE-Video datasets as well as a new dataset of instructional videos manually annotated with approximate 6D object trajectories. We demonstrate significant improvements over existing state-of-the-art RGB 6D pose estimation methods. Finally, we show that the 6D object motion estimated from Internet videos can be transferred to a 7-axis robotic manipulator both in a virtual simulator as well as in a real world set-up. We also successfully apply our method to egocentric videos taken from the EPIC-KITCHENS dataset, demonstrating potential for Embodied AI applications.

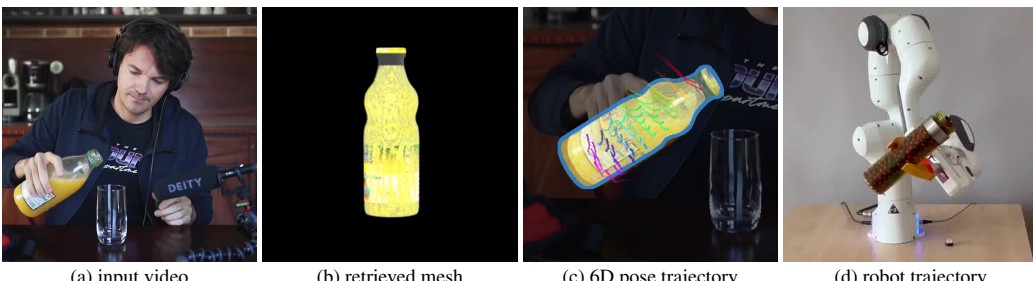

(a) input video      (b) retrieved mesh      (c) 6D pose trajectory      (d) robot trajectory

Figure 1: **Robotic manipulation guided by an instructional video.** (a) Given an instructional video from the Internet, our approach: (b) retrieves a visually similar mesh for the manipulated object from a large mesh database, (c) estimates the approximate 6D pose trajectory of the object across video frames, and (d) transfers the object trajectory onto a 7-axis robotic manipulator.

## 1 Introduction

In this work, we make a step towards learning robotic object manipulation skills from Internet instructional videos, as illustrated in Figure 1. This is a challenging problem as Internet instructional

---

[*]Equal contribution.

videos present uncontrolled capturing conditions where people manipulate a large variety of objects often in a dynamic fashion. While humans can easily learn even complex manipulation tasks from one or more instructional videos, building machines with similar cognitive and sensory-motor capabilities remains a major open challenge. Yet, such capabilities would enable tapping into millions of instructional videos available online (Miech et al., 2019).

Current methods for learning object manipulation skills from instructional videos have focused on learning manipulation for stick-like objects (e.g., hammer, scythe, or spade) (Li et al., 2022; Zorina et al., 2021), require tens or even hundreds of training examples captured from multiple viewpoints (Sermanet et al., 2018) or focus on recovering a static pose of the manipulated object (Patel et al., 2022; Wu et al., 2024). In contrast, we leverage the recent progress in 6D object pose estimation and object tracking, and develop a novel method for 6D object pose estimation that allows for extracting temporally consistent 6D object trajectories from uncontrolled Internet videos without any prior knowledge about the object itself. Despite being simple and training-free, the method can extract object trajectories that can be transferred to a robotic manipulator, making a significant step towards large-scale learning of robotic manipulation skills from Internet's instructional videos.

In detail, we make the following contributions:

- First, we develop a method that estimates the 6D pose of any object in an image without prior knowledge of the object itself. The method works by (i) retrieving a CAD model similar to the object in the image from a large-scale database, (ii) aligning the retrieved CAD model with the object, and (iii) grounding the object scale with respect to the scene.

- Second, we extract smooth 6D object trajectories from Internet videos by carefully tracking the detected objects across video frames. The extracted 6D object trajectories are then retargeted via trajectory optimization into the configuration space of a robotic manipulator.

- Third, we thoroughly evaluate and ablate our 6D pose estimation method on YCB-V and HOPE-Video datasets as well as a new dataset of instructional videos manually annotated with approximate 6D object trajectories. We demonstrate significant improvements over existing state-of-the-art RGB 6D pose estimation methods.

- Finally, we show that the 6D object motion estimated from Internet videos can be transferred to a 7-axis robotic manipulator both in a virtual simulator as well as in a real world robotic setup. Additionally, we apply our method to egocentric videos taken from the EPIC-KITCHENS dataset, showing our method can be useful in various Embodied AI applications.

## 2 RELATED WORK

**Object instance detection and mesh retrieval.** Image based pose estimation methods are usually decomposed into successive steps: object instance detection, often paired with mesh identification, followed by pose estimation. When the depicted objects all belong to a relatively small dataset (Xiang et al., 2018; Hodan et al., 2017; Tyree et al., 2022), most methods train an off-the-shelf 2D object detector, such as FasterRCNN Ren et al. (2016); Li et al. (2020), MaskRCNN (He et al., 2017; Labbé et al., 2020) or YOLOV8 (Jocher et al., 2023; Wang et al., 2021), using synthetic training data (Denninger et al., 2023). These supervised detectors achieve very high performance on standard benchmarks (Hodan et al., 2024) but are limited by the need of retraining for each new object instance. More recently, open-ended object detectors have appeared thanks to the advent of powerful foundation models (Ke et al., 2023; Oquab et al., 2024). For a given set of textured meshes, CNOS (Nguyen et al., 2023) proposes an onboarding phase where the known meshes are rendered across a set of viewpoints that are encoded with powerful image features such as (Oquab et al., 2024). At run time, candidate objects are segmented in the input RGB image using an object segmenter, such as SAM (Kirillov et al., 2023; Zhao et al., 2023), and encoded using the same image features. The depicted object instance is then matched to one of the known meshes whose image encoding is the most similar with the instance features across views. Recent advancements include reducing the number of false detections (Lu et al., 2024) by improving the instance segmentation (Nguyen et al., 2023) using (Liu et al., 2024b), or developing better view-level feature aggregation techniques such as Foreground Feature Averaging (Kotar et al., 2023).

**Model-based 6D pose estimation.** Recent progress in deep-learning based 6D pose estimation has been focused on increasing precision of alignment-based methods (Wang et al., 2021; Labbé et al.,

2020). This progress was made possible by the availability of large scale object databases (Downs et al., 2022; Deitke et al., 2023), procedural rendering software to produce synthetic training data (Denninger et al., 2023), and well-maintained public test datasets and benchmarks (Hodan et al., 2020). The recent results of the BOP challenge indicate that the metrics on 6D localization of objects known during training are starting to saturate (Hodan et al., 2024). The community effort has now shifted toward the development of models able to generalize to objects unseen during training (Zhao et al., 2022; Labbé et al., 2022; Nguyen et al., 2024; Örnek et al., 2024) which are often decomposed in coarse template matching (Nguyen et al., 2022) followed by iterative refinement (Hai et al., 2023). However, these model-based methods tend to lack robustness when the retrieved mesh differs too much from the object depicted in the input image, which limits their application in less controlled, "in the wild" settings. In this work, we aim to overcome this key limitation.

**6D pose estimation in the wild.** For uncontrolled setups, such as in Internet videos, one typically does not have access to exact models of the depicted objects. *Category-level* pose estimation methods drop the requirement for exact models by assuming the existence of a canonical reference frame among a class of objects (e.g., mugs or cars). Some of the works make simplifying assumptions by constraining the object rotation along the gravity direction, which is reasonable for large items such as furniture or vehicles (Papon & Schoeler, 2015; Braun et al., 2016). For 6D category pose estimation, the normalized object coordinate space representation is a popular choice (Brachmann et al., 2014; Wang et al., 2019; Xiao et al., 2022). In contrast, model-free pose estimation methods make no assumption on the type of objects present in the scene but rely either on a onboarding procedure where several reference images of the target object are captured together with the corresponding metric camera poses (Sun et al., 2022; He et al., 2022a) (e.g., using ARKit) or require the use of RGBD images as input at run-time (Wen & Bekris, 2021; He et al., 2022b). While the model-free methods are versatile, the absence of a metric mesh or a volumetric model limits their use for applications in simulation or robotics. Finally, another approach is to jointly reconstruct the object shape and solve for the camera pose(s) (Irshad et al., 2022). Object reconstruction may be done during an onboarding phase using one image (Long et al., 2024; Nguyen et al., 2024) or multi-view reconstruction (Yariv et al., 2020; Wen et al., 2024). However, the onboarding images are often required to display the object in a static pose and from several different views ($> 20$), which makes these methods unsuitable for directly estimating object poses from in-the-wild Internet videos. An alternative to onboarding can be test-time adaptation (Shi et al., 2024) to address the novel shapes at test time. Some works (Hasson et al., 2019; Zhang et al., 2020; Cao et al., 2021; Wen et al., 2023) have considered jointly reconstructing and tracking of objects interacting with hands, but none of these have demonstrated transferring object motion from in-the-wild Internet videos to a robotic manipulator. In contrast, we propose to tackle uncontrolled Internet videos via an approach combining mesh retrieval, 6D pose alignment, object tracking and trajectory optimization to go from the input unconstrained Internet video to a motion imitating the fine-grained human manipulation on a robot.

**Learning robotic manipulation from video.** One strategy for learning robot manipulation skills from human demonstrations consists of extracting reward signals from the videos and using the reward to train a robotic policy using reinforcement learning (a good survey of different methods is in (McCarthy et al., 2024)). Examples include (Sermanet et al., 2018), which defines the reward using image features extracted with an embedding model trained with time-contrastive self-supervision. The model requires a relatively large number of real demonstrations (around 100). Other works operate from a single demonstration and leverage explicit 3D object representations and estimated object poses to define the rewards. For example, (Petrík et al., 2021) uses a differentiable renderer to estimate the pose of geometric primitives approximating the objects shapes, and (Zorina et al., 2021) estimates the pose of stick-like objects (hammer, scythe) using the method of Li et al. (2022) and physics-based temporal smoothing. Closest to ours is Patel et al. (2022) which considers any type of object with a known 3D model. They demonstrate results for positioning objects into a static pose given by a video. In contrast, we do not assume the exact 3D model to be known and we show results for a complete task (e.g., pouring).

## 3 APPROACH

We first in Section 3.1 present our method for computing 6D poses for objects without knowing their exact 3D models. Then, in Section 3.2, we describe how we extract smooth 6D object trajectories from Internet videos and transfer them to a real robotic manipulator.

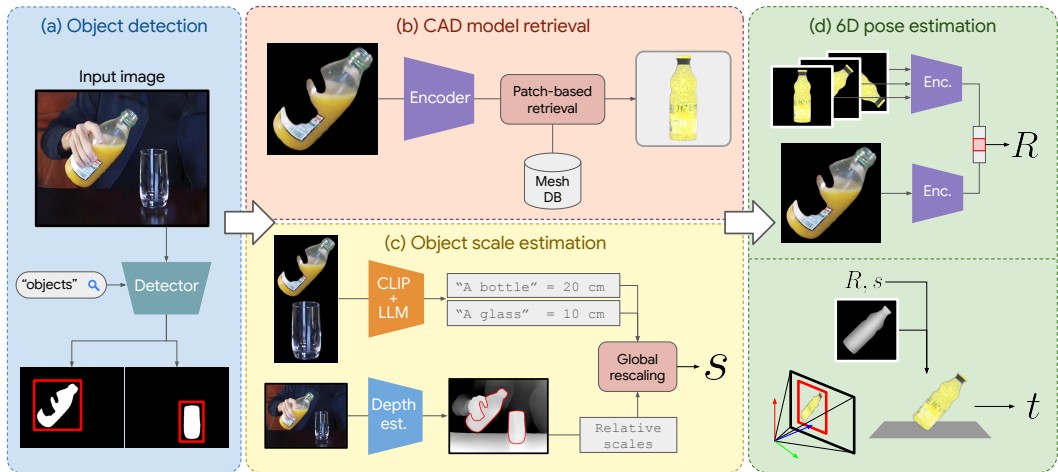

Figure 2: **Overview of our 6D object pose estimation without a known 3D mesh (Sec. 3.1).** Given an input RGB image, our method: (a) detects and segments objects present in the image, (b) retrieves similar meshes from a large-scale object database via patch-based retrieval, (c) estimates the absolute scale of depicted objects in the scene via LLM-based re-scaling, and (d) estimates the camera-to-object rotation $R$ and translation $t$ via alignment of the retrieved (approximate) mesh.

## 3.1 6D OBJECT POSE ESTIMATION WITHOUT A KNOWN 3D MESH

Given an image or a video frame, our goal is to detect and estimate the 6D pose of objects in the image. We wish to predict the object poses for in-the-wild images and videos without any prior knowledge of the objects themselves. Therefore, in contrast to the standard 6D pose estimation methods, our main challenge is the lack of a known CAD model of the object. Further, as the CAD model is unknown, the size of the object in the image is also unknown. The object can appear small in the image because it is small, or it is large but far from the camera.

To tackle these challenges, we present a novel method that **(a)** after detection and segmentation of the object in the image using an open-set detector, **(b)** retrieves a CAD model semantically and geometrically similar to the object in the image from a large database, **(c)** computes the object scale by estimating the monocular depth and grounding it to the real world using a large language model, and **(d)** estimates the 6D pose of the object in the image by aligning it with the retrieved CAD model. The overview of our approach is shown in Figure 2, and each of the steps is described below.

**Object detection and segmentation.** We use an open-set object detector with an image segmenter to generate high-quality object proposals and masks for objects present in the input image. Additional details can be found in Appendix A.1.

**CAD model retrieval.** Given an image or a video frame, our goal is to obtain CAD models of salient objects present in the image. While single-view 3D reconstruction methods (Wang et al., 2024; Szymanowicz et al., 2024; Liu et al., 2023) offer possible solution, we found them unsatisfiable for our task of 6D pose detection and tracking in the wild. Instead, for each object in the image, we retrieve the most similar CAD model from a large-scale reference database composed of Objaverse-LVIS (Deitke et al., 2023) and Google Scanned Objects (Downs et al., 2022).

The key question is how to perform retrieval of the most similar CAD model for an object in the input RGB image. This is hard as the database typically does not contain the exact CAD model for the depicted object. Hence, the goal is to retrieve another model instance from the same category (e.g., another similarly-looking mug). To perform such "category-level" retrieval, we use pre-trained *foundation* image features, as they are invariant to appearance differences between real images and CAD model renders. In detail, we represent each CAD model in the database as a single feature descriptor obtained by foreground feature averaging (FFA) (Kotar et al., 2023) of patch-level feature maps (Oquab et al., 2024) obtained from multiple rendered views of the given CAD model. For each proposal generated by the object detector and segmenter, we then extract the same FFA feature descriptor and retrieve the most similar CAD model from the database.

**6D object pose estimation via category-level view alignment.** Given the retrieved CAD model with its associated object mask in the input image, the goal is to estimate rotation $\mathbf{R} \in SO(3)$ and translation $\mathbf{t} \in \mathbb{R}^3$ of the object in camera coordinate space. We assume a pinhole camera model with focal length $f$, principal point $c$, and an approximate scale of the model. In Appendix A.2, we discuss how these assumptions can be relaxed.

First, we estimate the object's rotation via view alignment. In particular, we proceed as follows: we crop, mask, and rescale the image according to the object's mask and compute its patch features (Oquab et al., 2024), here denoted as $\{\mathbf{p}_k^{\text{query}}\}_k$ where $k$ is the patch index. Then, we compute the dot product with the patch features $\{\mathbf{p}_k^{\text{CAD}}(\mathbf{R})\}_k$ of a view of the CAD model rendered in rotation $\mathbf{R}$ and select the best rotation $\mathbf{R}^*$ maximizing the dot product averaged across all the $N$ patches:

$$\mathbf{R}^* = \arg\max_{\mathbf{R}} \frac{1}{N} \sum_{k=1}^{N} \langle \mathbf{p}_k^{\text{query}}, \mathbf{p}_k^{\text{CAD}}(\mathbf{R}) \rangle. \tag{1}$$

We use views rendered with rotations sampled using the viewpoint sampler of Alexa (2022).

The translation $\mathbf{t}$ is computed in two steps. First, the $z$ (depth) coordinate is computed by rescaling the CAD model's width, height, and depth $[o_w, o_h, o_d]$ in the rendered coordinate space $[\mathbf{R}^* | \mathbf{0}]$ by the object's bounding box width $b_w$ and height $b_h$ as :

$$\mathbf{t}_z = \frac{1}{2} \left( f \frac{o_w}{b_w} + f \frac{o_h}{b_h} \right). \tag{2}$$

Then, the $x$ and $y$ coordinates are computed using the bounding box center $[b_x, b_y]$ and the camera intrinsics as: $\mathbf{t}_x = (b_x - c_x)\mathbf{t}_z/f$ and $\mathbf{t}_y = (b_y - c_y)\mathbf{t}_z/f$. If the intrinsics of the camera are unknown, we observed that using common intrinsic values (equal to the field of view of 53 degrees) yields good results up to the absolute scale of the scene. The detailed discussion of this topic is in Appendix A.2. Determining the absolute scale is discussed next.

**Object scale estimation via monocular depth prediction and LLM.** Obtaining correct 6D pose of an object requires knowledge of its absolute scale. However, we cannot rely on the arbitrary scales of the CAD models found in our database. For example, the database models can have significantly different scales (metric or imperial, millimeters or meters), which makes it impossible to automatically use them for pose estimation tasks. Although one can roughly estimate the object scales from point clouds obtained by back-projection using a depth map, for in-the-wild scenarios, such as in Internet videos, the depth map is often not available. Instead, we develop a scale estimation method requiring only single-view RGB input, that takes advantage of a depth map estimated using a neural network and information about common object scales obtained from a large language model (LLM).

While it is impossible to predict a metrically accurate depth map from a single image, especially if the camera intrinsic parameters are not known, we observe that monocular depth predictors can be used to obtain relative scales of the objects even with unknown intrinsic parameters. We assign relative scale $r_i$ to every object $i$ as the distance between the furthest object point and object center along object's principal axis as computed from the point cloud obtained from the monocular depth predictor (Bhat et al., 2023), while using the common intrinsic priors (see the second paragraph in Appendix A.2). To obtain the metric scales $m_i$, we first ask GPT-4 (OpenAI, 2024) to generate a list of common object text descriptions with their typical size in meters. Then, we classify the depicted object using the dot product between CLIP (Cherti et al., 2023) features of the object's image and the text descriptions generated by GPT-4. We retrieve the corresponding scale as the scale of the best matching text descriptions as measured by the dot product. Note, that our set of generated text descriptions is not tailored to any specific scene type. Instead, it is designed to encompass a wide range of everyday objects, therefore it can be generated offline. Moreover, the set can be easily adapted to specific use cases by generating new descriptions using a modified prompt, if needed. Finally, we use the metric scales $m_i$ to globally rescale the relative scales $r_i$ obtained from the depth map. In this way, we keep the estimated ratios between the object scales, but employ the metric information obtained using the LLM. More details about the scale estimation method, including the outlier rejection procedure, can be found in Appendix A.3. Note that our approach aggregates scale estimates from multiple objects in the scene in a robust manner, which makes our method tolerant to incorrect scale estimates of some of the objects within the scene.

## 3.2 Robotic Object Manipulation from Internet Videos

In this section we describe how we obtain temporally consistent 6D pose trajectories from Internet videos and how we use those trajectories to drive a robotic manipulator. This problem is challenging as the input videos often depict fine-grained dynamic object motions (think person trying to empty a ketchup bottle). To address this issue we design an approach that outputs a smooth 6D motion in the configuration space of the robot by leveraging (i) approximate 6D object pose estimates obtained by the approach described in Section 3.1 with (ii) recent advances in accurate 2D feature point tracking in video (Karaev et al., 2024) and (iii) trajectory optimization techniques in robotics.

**Pose tracking for recovering object trajectories from Internet videos.** The category-level alignment approach described in Section 3.1 predicts a discrete set of rotations. Also, the alignment cannot recover the relative inter-frame rotation of cylindrically symmetric objects such as cups, bottles, or bowls due to the different textures of the real and the retrieved 3D objects. To produce smooth trajectories with correct inter-frame rotations that can be transferred to a robotic manipulator, we track the object across video frames using a dense point video tracker (Karaev et al., 2024). The tracking is initialized by sampling 3D points $\boldsymbol{p}_C \in \mathbb{R}^3$ on the retrieved object and projecting them into a video frame using the object pose estimate obtained using our 6D pose estimation approach, resulting in a set of 2D image points $\boldsymbol{p}_I \in \mathbb{R}^2$. The points $\boldsymbol{p}_I$ are tracked across video frames to produce a set of correspondences $\{(\boldsymbol{p}_C, \boldsymbol{p}_I^j)\}_p$ where $j$ is the index of a video frame. Assuming a static camera, the temporally refined 6D pose of the object is computed for each frame $j$ using the PnP algorithm (Lepetit et al., 2009) on the set of 3D-2D correspondences $\{(\boldsymbol{p}_C, \boldsymbol{p}_I^j)\}_p$. The temporally refined poses obtained from this procedure result in smoother trajectories with fewer errors for video frames with partial occlusions. Note that for moving cameras, camera movement correction can be made by estimating the relative camera poses from the static video background using readily available software such as COLMAP (Schönberger & Frahm, 2016). In this paper, we focus on object motion relative to the camera and thus do not explicitly estimate the camera's relative poses in the video.

**Retargeting trajectories to a robotic manipulator.** The temporally refined sequence of poses of the manipulated object is then used as a target to compute a trajectory in the configuration space of the robot imitating the object 6D trajectory extracted from the video. We address this problem using trajectory optimization as described next. To express the sequence of estimated object poses in the robot frame of reference, we manually locate the camera in the robot space and recompute the object poses estimated in the camera frame into the robot frame. We express the pose of the object with respect to the robot at time $t$ using transformation $\boldsymbol{T}_t$ that captures both the rotation and translation of the object in a homogeneous representation, as explained in (Sola et al., 2018). To obtain the robot trajectory, we use trajectory optimization (Jallet et al., 2023) to solve the following optimization problem:

$$\boldsymbol{\tau}_0, \dots \boldsymbol{\tau}_N = \underset{\boldsymbol{\tau}_0, \dots \boldsymbol{\tau}_N}{\arg\min} \sum_{t=0}^{N-1} w_d \log(\boldsymbol{T}_t^{-1} \tilde{\boldsymbol{T}}_t) + w_{\dot{q}} \left\| \dot{\boldsymbol{q}} \right\|^2 + w_\tau \left\| \boldsymbol{\tau} \right\|^2 , \quad (3)$$

where $\boldsymbol{\tau}$ is the torque that controls the robot joints; the first cost term weighted by $w_d$ penalizes SE(3) distance via SE(3) $\log$ function (Sola et al., 2018) between $\boldsymbol{T}_t$, pose of the object at time $t$ held by the robot, and $\tilde{\boldsymbol{T}}_t$, the pose of the object estimated from the video; the second and third terms of the cost regularize the velocity $\dot{\boldsymbol{q}}$ and torque $\boldsymbol{\tau}$ of the robot to prevent redundant motion of the robot. Optimization is solved using torque control actions to achieve a smooth trajectory in the robot configuration space. See Appendix A.6 for more details.

## 4 Experiments

In this section, we first describe the datasets and metrics used. Then we present a comparison of our method with state-of-the-art 6D pose estimation methods on the YCB-V and HOPE-Video datasets and we ablate the key components of our approach. Finally, we evaluate the proposed method on challenging Internet videos and demonstrate that the estimated 6D object trajectories can be imitated by a robotic manipulator. Additionally, in Appendix C.5 we provide quantitative results on DTTD2 (Huang et al., 2023) dataset.

Table 1: **Comparison with the state-of-the-art on the YCB-V and HOPE-Video datasets.** The 6D pose estimation results of our approach compared to MegaPose (Labbé et al., 2022), Giga-Pose (Nguyen et al., 2024), and FoundPose (Örnek et al., 2024). Our method outperforms all other methods in all the metrics.

| Method | YCB-V | | | | HOPE-Video | | | |
|---|---|---|---|---|---|---|---|---|
| | AR↑ | AR$_{CoU}$↑ | AR$_{CH}$↑ | AR$_{pCH}$↑ | AR↑ | AR$_{CoU}$↑ | AR$_{CH}$↑ | AR$_{pCH}$↑ |
| MegaPose (coarse) | 23.75 | 10.08 | 10.65 | 50.53 | 31.77 | 9.96 | 6.87 | 78.50 |
| MegaPose | 25.76 | 14.01 | 11.91 | 51.37 | 33.03 | 13.07 | 6.38 | 79.64 |
| GigaPose | 29.18 | 11.90 | 9.20 | 66.45 | 23.12 | 4.15 | 4.90 | 60.30 |
| FoundPose | 42.95 | 35.40 | 15.69 | 77.75 | 42.30 | 31.18 | 9.58 | 86.13 |
| Ours | **49.86** | **45.20** | **18.53** | **85.83** | **45.98** | **39.21** | **10.72** | **88.01** |

Table 2: **Influence of different model retrieval methods on 6D pose estimation.**

| Retrieval | AR↑ | AR$_{CoU}$↑ | AR$_{CH}$↑ | AR$_{pCH}$↑ |
|---|---|---|---|---|
| (a) OpenShape | 34.51 | 20.25 | 10.59 | 72.69 |
| (b) Ours (`CLS`) | 45.05 | 40.93 | 16.16 | 78.06 |
| (c) Ours | **49.86** | **45.20** | **18.53** | **85.83** |
| (d) Oracle | 62.93 | 51.99 | 45.95 | 90.85 |

Table 3: **Effects of scale estimation method on the 6D pose estimation.**

| Scale Estimator | AR$_{CH}$[†]↑ |
|---|---|
| Constant | 11.89 |
| Ours | **18.53** |
| Oracle | 45.95 |

[†] Other metrics are scale-independent.

## 4.1 Evaluation of 6D Pose Estimation on Standard Datasets

**Datasets and metrics.** We evaluate our method on YCB-V (Xiang et al., 2018) and HOPE-Video (Lin et al., 2021). The datasets feature cluttered scenes with partially occluded objects, resembling our target Internet videos. Both datasets contain full 6D pose annotation with the known object meshes available, yet, in our experiments, we utilize the meshes retrieved from Objaverse-LVIS (Deitke et al., 2023) and Google Scanned Objects (Downs et al., 2022) datasets instead. This simulates the in-the-wild scenario where no *ground-truth* meshes are available.

For evaluation, we follow the BOP-inspired protocol, where **average recall** (AR) is estimated for every metric. We report the standard Chamfer distance (CH) metric. However, because this metric is highly dependent on the correct CAD model scale which is unavailable in our scenario, we also report projection-based metrics, i.e., projected Chamfer distance (pCH) and complement over union (CoU). We also report mean AR across all metrics. For comparison of proposal generation methods, we also report average precision. See details of the metrics in Appendix B.

**Comparison with the state of the art.** We compared our RGB-only approach against state-of-the-art RGB-only methods from the BOP challenge's "unseen-objects" category, as our target YouTube videos lack depth information. The top open-source methods compared were GigaPose (Nguyen et al., 2024), FoundPose (Örnek et al., 2024), and MegaPose (Labbé et al., 2022). We always use retrieved meshes and scales estimates instead of the ground truth meshes for a fair comparison. The results of the methods on the YCB-V and HOPE-Video datasets are shown in Table 1. The table shows our method significantly outperforms all three methods, and impressively, our method achieves that without any in-domain training or fine-tuning. This indicates that MegaPose, GigaPose and FoundPose, designed for 6D pose estimation with precise and known meshes, do not generalize well to scenarios with unknown or imprecise meshes and are unsuitable for the in-the-wild task.

### 4.1.1 Ablations of Key Components

**CAD model retrieval.** We compare our method for CAD model retrieval to the OpenShape (Liu et al., 2024a) baseline on the YCB-V dataset in Table 2. OpenShape is a multi-modal model that aligns point clouds into CLIP space using PointBERT (Yu et al., 2022). While OpenShape works well when the ground truth models are present in the database, in our scenario, when the exact models are unavailable, our method **(c)** outperforms OpenShape **(a)** by roughly 44% as measured by final pose estimation quality. As the upper bound for CAD model retrieval, we also show our method in the scenario when the exact meshes of the objects depicted in the image and their scales are known **(d)**. Additionally, we compare our choice of using the `FFA` descriptor with the per-view

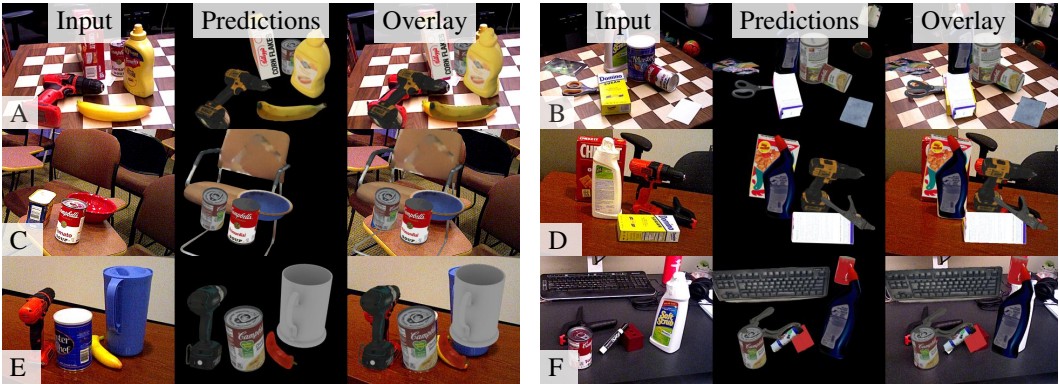

Figure 3: **Qualitative results of our method on the YCB-V dataset.** On the YCB-V dataset our method is able to detect objects that are not part of the dataset (such as keyboard in image F and chair in image C), which highlights the benefit of our method of not being restricted to the ground truth meshes.

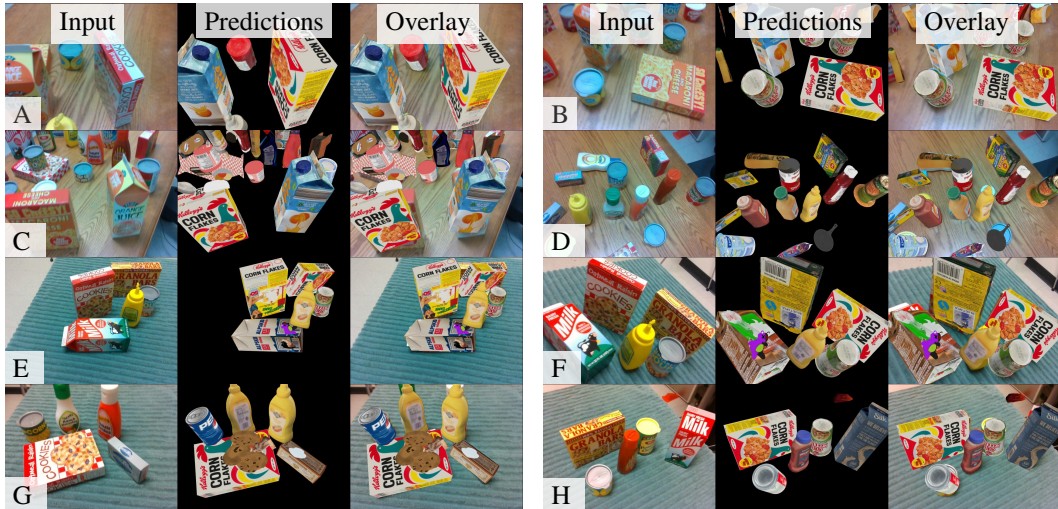

Figure 4: **Qualitative results of our method on the HOPE-Video dataset.** On the HOPE-Video dataset our method reconstructs complex scenes with multiple objects while respecting partial occlusions.

averaged `CLS` token descriptor. We show that foreground feature averaging **(c)** outperforms the `CLS` token **(b)** substantially. We provide further ablations on the descriptor choice in Appendix C.1.

**Scale estimation.** We compare our object scale estimation method to the constant scale baseline of 10cm on the YCB-V dataset in Table 3. Even though estimating scale from a single image is an ill-posed problem, our method that uses a scale prior obtained from a large language model improves the average Chamfer distance recall by 55% over the constant model scaling. This shows the LLM brings insightful information about the object sizes.

### 4.1.2 QUALITATIVE RESULTS OF 6D POSE ESTIMATION

The qualitative results on YCB-V images are shown in Figure 3 and qualitative results on HOPE-Video are shown in Figure 4. The successful examples in the figure demonstrate the capability of the proposed method, that is, being able to retrieve an object in the same category and to align its pose despite visual differences. For example, consider the drill in the Figure 3-A: our method retrieves another type of drill whose texture differs from the texture in the query image, but the retrieved CAD model is still aligned with the input query image. A similar example is shown in Figure 4-A, where our method aligns meshes with textures different from the objects depicted in the input image. An added benefit of our method is it's capability to align objects that are outside of the training dataset.

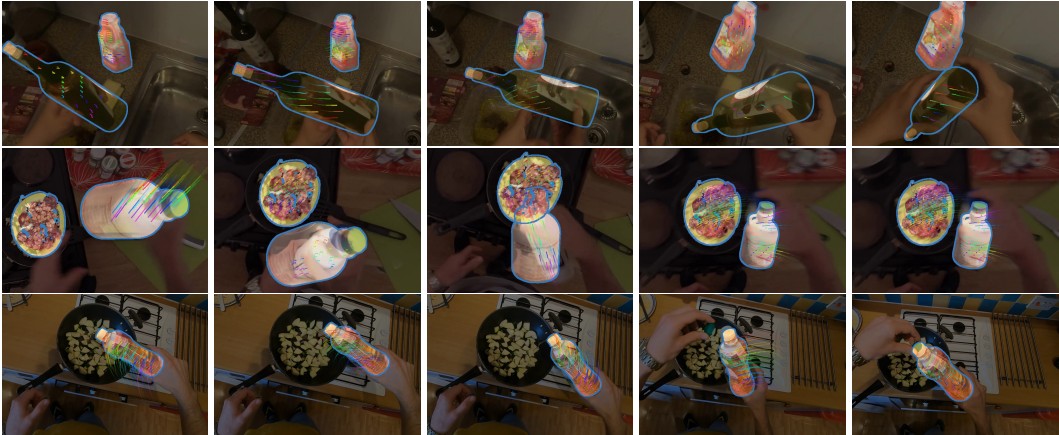

Figure 5: **Qualitative results of our method on the EPIC-Kitchens dataset.** Our method is able to detect and align objects from an egocentric point of view. See Figures 8 and 9 for additional examples.

In Figure 3-C our method retrieves a chair and in Figure 3-F a computer keyboard, both of which are not in the set of ground truth objects of the YCB-V dataset.

**Failure modes.** Our method has two main failure modes: (i) failure to retrieve a similar mesh and (ii) failure to estimate the correct scale. The example of the first failure mode is shown in Figure 4-C, where instead of the box (top left) a red and white checkerboard object is retrieved and as a consequence the pose estimation fails. Another failure mode is shown in Figure 4-E where improper scale estimation leads to incorrect scene occlusions.

## 4.2 VIDEO RESULTS AND APPLICATIONS

We show quantitative and qualitative results of our full 6D pose detection and tracking method on Internet videos as well as on egocentric videos from the EPIC-KITCHENS dataset (Damen et al., 2022). We also show an application of the extracted object manipulation trajectories on a real robot. More results are available in the appendix and the **supplementary video**.

**Quantitative evaluation.** For the quantitative evaluation of object tracking in-the-wild, we manually annotated an approximate 6D pose of human interacted objects in 32 "in-the-wild" videos containing a total of 6947 frames. To annotate the ground truth poses, the annotator selected an object from the top Objaverse retrievals and then manually aligned it to visually match each frame of the video. The results of our approach are compared to several state-of-the-art methods are shown in Table 4. Our method consistently improves upon the previous state-of-the-art. More details on the evaluation metrics are in Appendix B.2.

**Qualitative results on Internet and egocentric videos.** For a qualitative evaluation of the proposed method in the wild, we consider instructional videos, e.g., cooking videos from the Internet and EPIC-KITCHENS (Damen et al., 2022) dataset. The resulting 6D object trajectories obtained from videos from the EPIC-KITCHENS dataset are shown in Figure 5 and Figures 8 and 9 in the

Table 4: **Comparison with state-of-the-art 6D object estimation methods on "in the wild" videos.** The 6D object tracking results of our approach are compared to MegaPose (Labbé et al., 2022), GigaPose (Nguyen et al., 2024), and FoundPose (Örnek et al., 2024). Our method outperforms these baselines in all metrics.

|  | MegaPose (coarse) | MegaPose | FoundPose | Ours | Ours (refined) |
|---|---|---|---|---|---|
| Relative rotation ↓ | 6.866 | 6.822 | 7.736 | **1.719** | **1.205** |
| Relative projected translation ↓ | 0.107 | 0.161 | 0.405 | **0.091** | **0.086** |

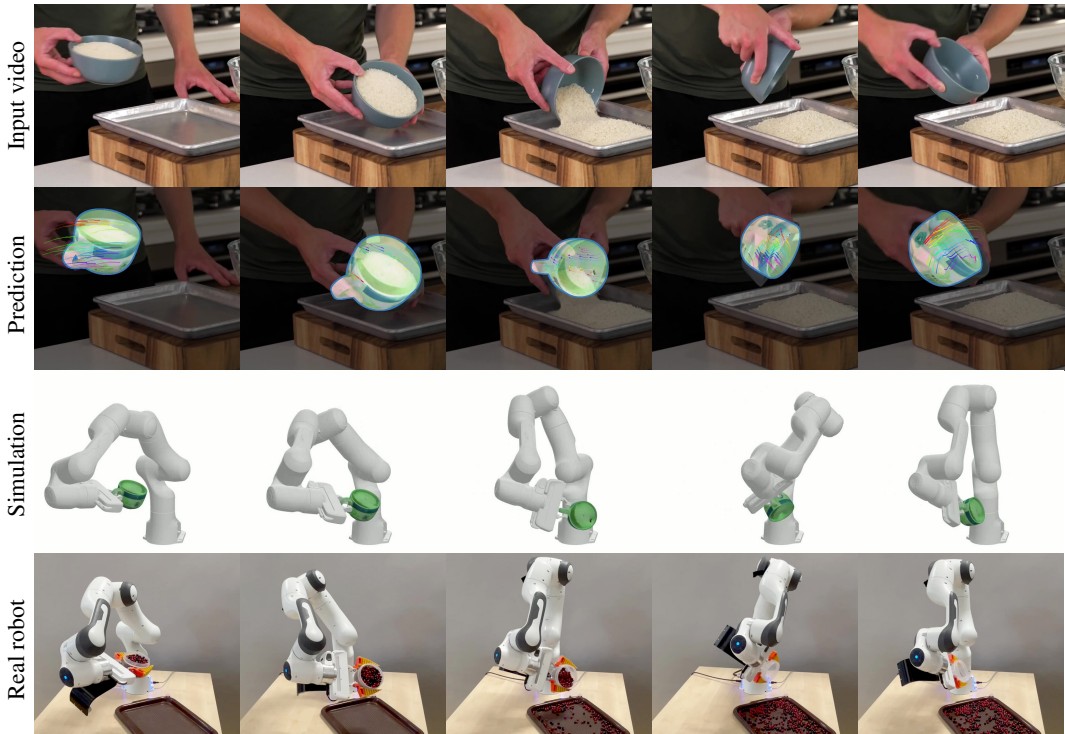

Figure 6: **The proposed approach applied on an Internet video.** Top row: several input frames. Second row: alignment obtained by our method together with tracked 2D-3D correspondences shown as colored tracks. Third row: Temporally smooth poses are used to compute robot motion via trajectory optimization. Bottom row: video of a real robot execution. **Please see additional results in the Appendix D and the supplementary video.**

appendix. Please note, our method can detect and track multiple objects. We use the same approach for the Internet instructional videos and then perform experiments on a real robot, as described in the following paragraph.

**Application: Imitating object motion from Internet videos.** The temporally smooth 6D poses are used to optimize the trajectory of the Franka Emika Panda robot. The optimized trajectory can then be executed in simulation or on a real robot. The proposed approach successfully replicates actions shown in instructional videos, as shown in Figure 6. Additional results are shown in Figure 10 and Figure 11 in Appendix D. For full videos of the real-world robot manipulation, please see the supplementary video. Please note how in the video the robot imitates the extracted subtle human motions that would be otherwise hard to program, e.g., shaking the jug before pouring.

**Limitations.** Two main limitations persist: (i) if the quality of object segmentation is low or the object is significantly occluded, the retrieval and scale estimation may not be successful; and (ii) the mesh retrieval sometimes selects the most semantically similar object, which however, is not always geometrically similar to the object in the video, which can lead to inaccuracies in pose estimation.

## 5 CONCLUSION

We present an approach for category-level 6D object alignment for in-the-wild images and videos. We show that a similar mesh can be retrieved from a large-scale database of CAD meshes given the input query image, and can be then aligned to the query image despite the visual dissimilarities. We demonstrate quantitatively on two BOP datasets and a new dataset of "in-the-wild" instructional videos that our method outperforms the state-of-the-art 6D pose estimation baselines. Qualitatively, we show an application of the proposed method to guiding robotic manipulation using an instructional video, making a step towards large-scale learning of object manipulation skills from Internet's instructional videos.

ACKNOWLEDGMENTS

This work was partly supported by the Ministry of Education, Youth and Sports of the Czech Republic through the e-INFRA CZ (ID:90254), and by the European Union's Horizon Europe projects AGIMUS (No. 101070165), euROBIN (No. 101070596), and ERC FRONTIER (No. 101097822). Views and opinions expressed are however those of the author(s) only and do not necessarily reflect those of the European Union or the European Commission. Neither the European Union nor the European Commission can be held responsible for them. Georgy Ponimatkin and Martin Cífka were also partly supported by Grant Agency of the Czech Technical University in Prague under allocations SGS25/156/OHK3/3T/13 (GP) and SGS25/152/OHK3/3T/13 (MC).

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

APPENDIX

In this Appendix, we provide additional technical information related to our method. In Section A, we describe implementation details of our method. The details on CAD model retrieval are described in Section A.1. In Section A.2, we provide details on the rotation estimation part of our pipeline together with a discussion on how camera focal length priors can be used in case of unknown focal length. Then, in Section A.3, we describe the technical details of the scale estimation part of our method. In Section A.4, we provide more technical details on object tracking. Section A.5 then provides details on the runtime of our method. Section A.6 provides details on motion retargeting to a robotic manipulator. Section B provides details on metrics used in the evaluation of our method. Section C describes additional abblation studies of our method. Finally, in Section C.5 and Section D, we provide additional quantitative and qualitative results obtained by our method.

# A IMPLEMENTATION DETAILS

## A.1 CAD MODEL RETRIEVAL DETAILS

Our model database contains approximately ∼50,000 objects from Objaverse-LVIS (Deitke et al., 2023) and Google Scanned Objects (Downs et al., 2022) datasets. Each CAD model is rendered in $M = 600$ views at 420x420 pixels, with rotations sampled from SO(3) using viewpoint sampler (Alexa, 2022) from which 1024D `FFA` representation descriptor is computed. When constructing `FFA` mesh descriptor the averaged foreground patch token is computed from layer 22 of the DINOv2 model (Oquab et al., 2024) for each of the $M = 600$ rendered views of the CAD model. The resulting per-view foreground patch tokens are then aggregated to create per-object patch token. While such a representation may seem expensive to compute, it needs to be computed only once for each CAD model in the database.

To retrieve CAD models for a given image, we first generate proposals using GroundingDINO (Liu et al., 2024b) with prompt `"objects"` combined with SAM 2 (Ravi et al., 2024). Then, the retrieval process is done by matching a single 1024D FFA descriptor extracted from a query image to a database of ∼50,000 1024D descriptors (one descriptor for one object) by means of the dot product.

## A.2 6D OBJECT POSE ESTIMATION DETAILS

**Rotation estimation.** The rotation estimation part is built on the pre-trained DINOv2 ViT-L / 14 (Oquab et al., 2024) model. The rotation is estimated using spatial patch features from layer 22 of DINOv2 model by directly matching proposal image to $M = 600$ pre-rendered templates of the retrieved CAD model. The proposal image is cropped and resized to 420x420 pixels to match the resolution of pre-rendered templates, which gives feature maps of 30x30 tokens for matching.

**Camera focal length priors.** In general, for the complete 6D pose estimation, both the camera intrinsics and the exact CAD model with the correct absolute scale must be known. Knowing only one of these is insufficient. Therefore, when ground truth camera focal length is not available, we use the commonly used prior of the camera focal length $f = \sqrt{w^2 + h^2}$, where $w$ and $h$ are image width and height in pixels, to obtain an approximate point cloud. Our method tackles the problem when the exact CAD model with the ground truth scale is unavailable. Hence, by definition, there are infinitely many solutions (up to the unknown scale of the object) and defining absolute translation error is not meaningful both with and without known camera intrinsics. To get an approximate translation, our method estimates the absolute scales of objects in the scene by leveraging a prior knowledge available in an LLM as is described in Section 3.1 and, similarly, we use the commonly used prior of the camera focal length as stated above when the intrinsics are not available.

For robotic experiments, we rescale the predicted object trajectory so that the robotic arm can execute the motion. This rescaling would be necessary, even if both the camera intrinsics and the ground truth CAD model were known, as the robots and manipulated objects vary in size and the motion executed by a human in the video needs to be translated into the space of the robot. However, this task-specific scale refinement can be done automatically using, for example, a task specific reward,

e.g. to maximize the amount of poured liquid, in the aligned simulated environment as used, e.g., in Zorina et al. (2021).

## A.3 OBJECT SCALE ESTIMATION DETAILS

**LLM scale database generation.** When obtaining metric scales using an LLM, we precompute a database of object text descriptions and their typical size in meters. To create the database, the keywords were generated by the GPT-4 (OpenAI, 2024) model by repeatedly querying the following prompt:

```
You help researchers to understand the 3D world around them.  You
need to generate 250 keywords for common real-world objects and you
also need to estimate the average object scale in that category by
considering the average largest dimension in meters.  The keyword
should be generated in the form of an article, followed by adjective
(if needed), and then by noun.

Examples:  a coffee mug, a computer mouse, a pan, a laptop.

In output, please ignore natural entities such as mountains, planets,
do not include imaginary creatures, and buildings.  Also, do not
consider niche objects such as diving equipment or rare scientific
equipment.

The output should be following this structure:

object_1:  scale_1

object_2:  scale_2

The input should be made only of this list.  The scales should
represent numbers in meters, but do not include units.
```

The scales obtained were accumulated and, in case of repeated occurrence, the median scale was taken, and saved in the database with CLIP feature vectors of the text descriptions. In total, we collected a list of 2200 objects, with scale values ranging from 1cm to 300m ("a cruise ship"), with 90% of the objects being under 2m. This makes it possible to use the proposed approach for the most common indoor scenarios, but possibly some outdoor scenarios as well. However, when a different environment is expected, the list of objects can be easily adapted by adjusting the LLM prompts to generate objects from this environment.

**LLM scale retrieval.** During inference, we extract CLIP features from the images of masked objects depicted in the input scene. Then, we retrieve the $K$ most similar text descriptions with the corresponding scales for each object using the extracted CLIP features, and aggregate the $K$ retrieved scales using the median into metric scale $m_i$ per object $i$.

**Relative scale estimation.** To obtain relative scales of objects in a scene, we use the depth estimation approach (Bhat et al., 2023) to obtain a depth map and compute a pointcloud for each object $i$ in the scene using its segmentation mask. We then identify the direction of the centered point cloud with the largest variance using the SVD decomposition and estimate the relative object scale $r_i$ for object $i$ as the size of the largest dimension of the point cloud.

**Global rescaling.** To combine the metric scale $m_i$ from the LLM with the relative scale $r_i$ from the depth map for object $i$, we first compute the ratio $\rho_i = m_i/r_i$ between the scales obtained using these two methods. Then, we take the median value of these ratios over all objects in the scene obtaining a single correction factor $\rho$ for the entire scene. Finally, we obtain the metric scale $s_i$ for each object $i$ by correcting the relative scale $r_i$ obtained from the automatic depth estimation using the scene correction factor $\rho$ as $s_i = r_i \, \rho$. The outcome is an estimate of a real-world scale $s_i$ (in meters) for each extracted object $i$.

Please note that for some objects with high size variation, the size can be difficult to estimate, e.g. a bicycle for a kid or an adult, or a real-sized car compared to a toy car. To cope with these issues, we use median aggregation during the global rescaling, making our method robust to outliers, i.e. tolerating large portions of incorrect scale estimates within the same scene.

### A.4 POSE TRACKING DETAILS

To track the object, we detect objects in the first frame of the input video using a strong open set object detector GroundingDINO (Liu et al., 2024b) to localize salient object candidates, and then track them in the video frames using SAM2 (Ravi et al., 2024). We then use our 6D pose estimation method on each frame to retrieve similar CAD model from the database and align them in each frame of the video. The highest confidence alignment is used to initialize pose tracking based on 3D-2D correspondences obtained using the 6D pose from the selected frame and 2D tracks from CoTracker (Karaev et al., 2024).

### A.5 COMPUTATIONAL RESOURCES AND RUNTIMES

All experiments were carried out on a cluster featuring nodes with 8x NVIDIA A100 (40 GB of VRAM per GPU), 2x 64-core AMD EPYC 7763 CPU, 1024 GB RAM. The total storage needed for this project was around 4TB. Each experiment on the YCB-V dataset takes around 6 hours on a single GPU and on the HOPE-Video dataset around 14 hours on a single GPU.

The method onboarding stage consists of pre-rendering the meshes and extracting the features. The Objaverse-LVIS database with roughly ∼50,000 objects can be rendered in 3 days on one 8-GPU node. The extraction of visual features for retrieval then takes around 3 days on one 8-GPU node as well, and takes around 200MB of disk space. In practice, we used the before mentioned cluster to parallelize and speed up the rendering and extraction processes significantly.

When running on a single image, the resulting runtime for detection, retrieval, and scale estimation is ∼2s per image. For the pose estimation part, the runtime is on average ∼0.2s per object with caching. However, when running on an instructional video, we run the detector only on the first frame and track the detected objects through the video using SAM 2 (Ravi et al., 2024), which can run in real-time. Then we run the retrieval and scale estimation, which can benefit from multi-frame prediction aggregation, but does not necessarily have to use all video frames, especially for longer videos. In practice, we used 30 frames throughout the video for the retrieval and scale estimation. In contrast to single-image estimation, we use CoTracker (Karaev et al., 2024) combined with PnP to extract smooth poses from the video, with run-time of ∼1s per frame.

### A.6 RETARGETING TRAJECTORIES TO A ROBOTIC MANIPULATOR

**Initial object pose replication.** To initiate the trajectory following process, the object needs to be placed in an initial configuration that corresponds to the starting pose of the video demonstration. We manually placed the object into the gripper (effectively fixing the transformation from the gripper to the object). The robot was then commanded to move the object to the initial pose, calculated as $T_{RC}T_{CO}^0$, where $T_{CO}^0$ is the object pose relative to the camera frame at the first frame of the video, as estimated by our method, and $T_{RC}$ is the fixed transformation from the robot base to the virtual camera frame, manually chosen to simulate a camera viewing the robot from the front with a $30°$ elevation relative to the gravity vector. Transformation $T_{RC}$ was manually defined and held constant across all demonstrations. This allows us to move the object to a pose visually similar to the initial state in the videos. However, in a practical robotic system, this manual initialization would be superseded by pre-existing grasp configurations or an automated grasping procedure (e.g., using a combination of motion planning and GraspIT (Miller & Allen, 2004) or GraspNet (Fang et al., 2020)).

**Trajectory following via optimization.** To achieve trajectory transfer, we represent the object's motion relative to its initial pose, $T_{CO}^0$. This yields a sequence of relative transformations, $(T_{CO}^0)^{-1}T_{CO}(t)$, that define the object's movement. A reference trajectory is then generated by applying these relative transformations to a chosen initial pose in the robot's task space in simulation or real life. Finally, trajectory optimization (Eq. 4) determines the joint torques necessary for the robot to imitate this reference trajectory. We employed the publicly available dynamic model for the Franka Emika Panda robot from the `example-robot-data` package (available via Conda and PyPI), which provides kinematic, geometric, and inertial parameters for each robot link. For forward dynamics computations, we utilized the `Pinocchio` library (Carpentier et al., 2019), also employed internally by the `Aligator` trajectory optimization package (Jallet et al., 2023). Given a sequence of joint torques, the dynamic model computes corresponding joint positions and velocities.

Forward kinematics, based on the kinematic model, then maps these joint states to the object's pose. Consequently, the optimized joint torques (Eq. 4) directly influence both the end-effector pose and the joint velocities.

# B  METRICS

## B.1  SINGLE FRAME METRICS

Our method is evaluated on three different metrics, **Complement over Union**, **Chamfer Distance** and **Projected Chamfer Distance**. Each of the metrics is described in more details below.

- **Complement over Union** (CoU) is defined as IoU between the rendered mask of the ground truth mesh under the ground truth pose and the predicted mesh under the predicted pose. This metric measures the visual alignment of the two masks as well as the silhouette appearance of the two objects.

- **Chamfer Distance** (CH) is a standard measure used to compare similarity of two pointclouds. This metric was chosen since the only way to compare quality of the pose estimate in 3D is via geometric alignment of the ground truth (GT) and predicted meshes under GT and predicted poses accordingly. This metric takes into account both pose and scale predictions and hence allows to see full performance of our model in 3D.

- **Projected Chamfer Distance** (pCH) is complementary to the full Chamfer distance since it allows to evaluate how well estimated poses reproject into the image, essentially marginalizing over the scale in the chamfer distance. Inspired by the MSPD (Maximum Symmetry-Aware Projection Distance) metric used in the BOP challenge, pCH is given by

$$\text{pCH} = \frac{1}{|M|} \sum_{x \in M} \min_{y \in \hat{M}} ||\pi(x, \boldsymbol{K}, \boldsymbol{T}) - \pi(y, \boldsymbol{K}, \hat{\boldsymbol{T}})||_2^2$$
$$+ \frac{1}{|\hat{M}|} \sum_{y \in \hat{M}} \min_{x \in M} ||\pi(x, \boldsymbol{K}, \boldsymbol{T}) - \pi(y, \boldsymbol{K}, \hat{\boldsymbol{T}})||_2^2 \tag{4}$$

  where $\boldsymbol{K}$ is camera intrinsics, $\boldsymbol{T}$ is a predicted pose, $\hat{\boldsymbol{T}}$ is ground truth pose, $M$ is a set of points sampled from the predicted mesh, $\hat{M}$ is a set of points sampled from the ground truth mesh, $x$ and $y$ are vertices of the meshes, and function $\pi$ projects 3D vertex into 2D pixel. The core idea of this metric is to evaluate how well the estimated pose visually aligns when projected onto the image plane. As the closest vertex is found for each vertex of other mesh, this metric can be used for non-identical meshes or for symmetric meshes. This approach allows us to assess alignment quality, even when the predicted scale is not exact, since the scale is factored out during the projection.

## B.2  TRACKING METRICS

To allow comparison of different methods, we collect 32 videos consisting of 15 instructional YouTube videos and 17 egocentric videos from the EPIC-KITCHENS (Damen et al., 2022) dataset. These videos contain altogether 6947 frames. We manually create the ground-truth annotations by visually aligning the semi-automatically retrieved mesh from the Objaverse dataset with the manipulated object in the video. When comparing the estimated object trajectories of different methods, we focus more on the relative object motion rather than precise 6D object trajectory. To this end, we use three metrics that measure spatial velocity error in rotation, projected translation, and depth. Specifically, we compute errors as the average error over all pairs of poses in video frames separated by a temporal distance $\delta$, where $\delta \in \Gamma$ and $\Gamma$ is a set containing 10 integer values linearly spaced from 1 to $\lfloor N/2 \rfloor$. Here, $N$ represents the total number of frames in the video, and $\lfloor \cdot \rfloor$ denotes the floor function, rounding down to the nearest integer. The overall errors are computed as:

$$e^{\text{rot}} = \frac{1}{|\Gamma|} \sum_{\delta \in \Gamma} \frac{1}{N - \delta} \sum_{i=1}^{N-\delta} \frac{1}{\delta} e_{i,i+\delta}^{\text{rot}} \,, \tag{5}$$

$$e^{\text{proj}} = \frac{1}{|\Gamma|} \sum_{\delta \in \Gamma} \frac{1}{N - \delta} \sum_{i=1}^{N-\delta} \frac{1}{\delta} e_{i,i+\delta}^{\text{proj}} \,, \tag{6}$$

$$e^{\text{depth}} = \frac{1}{|\Gamma|} \sum_{\delta \in \Gamma} \frac{1}{N - \delta} \sum_{i=1}^{N-\delta} \frac{1}{\delta}\, e^{\text{depth}}_{i,i+\delta}, \tag{7}$$

where $e^{\text{rot}}_{i,i+\delta}$, $e^{\text{proj}}_{i,i+\delta}$, and $e^{\text{depth}}_{i,i+\delta}$ are errors computed between frames $i$ and $i+\delta$ for rotation, projected translation and depth. These errors are divided by the time difference $\delta$ to obtain a relative error per frame. The inner sum computes the error across the $N - \delta$ frames of the whole video, which is then normalized by $N - \delta$ to obtain the average error. With the averaging over multiple time differences $\delta \in \Gamma$ (the outer sum), we penalize inconsistencies in velocities both locally ($\delta = 1$) and globally $\delta = \lfloor N/2 \rfloor$.

**Rotation error.** Given the estimated rotations $\boldsymbol{R}_i, \boldsymbol{R}_j$ and the ground-truth rotations $\hat{\boldsymbol{R}}_i, \hat{\boldsymbol{R}}_j$ for frames $i$ and $j$, we compute the rotation error $e^{\text{rot}}_{i,j}$ as the error of the ground-truth and the estimated rotation spatial velocities in the camera frame, expressed in degrees. In addition, we minimize the rotation error over the set of discretized ground-truth object symmetries $\mathcal{S}$, when the object symmetry axis is defined by the human annotator (if available). The error is computed as:

$$e^{\text{rot}}_{i,j} = \min_{\boldsymbol{S} \in \mathcal{S}} \left\| \log(\boldsymbol{R}_i \boldsymbol{R}_j^\top) - \log(\hat{\boldsymbol{R}}_i \boldsymbol{S} \hat{\boldsymbol{R}}_j^\top) \right\|, \tag{8}$$

where $\log$ is SO(3) logarithm (Sola et al., 2018). Note that because we compare spatial velocities when computing rotation error, having different meshes for ground-truth annotation and the estimation does not affect the rotation error. However, this is not necessarily true for projected translation and depth errors, as explained in the following paragraph.

**Projected translation error.** When computing projected translation error, we need to consider that we can have two different meshes for the ground-truth and the estimation, as their coordinate system origins do not necessarily have to project to the same 2D point in the image. As illustrated in Figure 7, the velocity of the projected object center can change differently, depending on the 3D position and the rotation of the object. We therefore recompute the translational part of the estimated object trajectory $\mathbf{t}$ to a new trajectory $\mathbf{t}^*$ that corresponds to the same object motion but with a changed origin of the object coordinate system. In the best-case scenario, where the ground-truth and estimated trajectory would be exactly the same up to the choice of the object origin, we would shift the estimated object origin so that it is the same 3D point as the origin of the ground-truth object when expressed in the camera coordinate frame. This is, however, impossible because the objects can have different velocities in different video frames, and also because of the ambiguity between the object's depth and scale. We, therefore, use the following approach instead: In each video frame $i$, we first compute a point that lies on the ray to the ground-truth object center, i.e. $\hat{\mathbf{t}}_i$, but having the same distance from the camera as the estimated object. Then, we express this point in the coordinate frame of the estimated object and compute the mean over all video frames, resulting in the new object origin $\mathbf{o}^*$:

$$\boldsymbol{o}^* = \frac{1}{N} \sum_{i=1}^{N} \boldsymbol{T}_i^{-1}\, \hat{\boldsymbol{t}}_i / \|\hat{\boldsymbol{t}}_i\| \cdot \|\boldsymbol{t}_i\|. \tag{9}$$

Finally, by changing the object coordinate system, we compute the new translations as:

$$\mathbf{t}_i^* = \boldsymbol{t}_i + \boldsymbol{R}_i \boldsymbol{o}^*. \tag{10}$$

Note that by using this approach, the positions of the mesh vertices do not change, but we can compensate for undesirable errors caused by having trajectories of two different meshes. In addition, we limit the change of object origin by half of the estimated object scale to avoid a significant unintentional error decrease in special cases of degenerate trajectories.

Given the ground-truth translations $\hat{\mathbf{t}}$ and the corrected translations $\mathbf{t}^*$, we now compute error of the projected translation $e^{\text{proj}}_{i,j}$ as the error between the spatial velocities of the projected object origins:

$$e^{\text{proj}}_{i,j} = \frac{100}{\sqrt{w^2 + h^2}} \left\| (\pi(\boldsymbol{t}_i^*, \boldsymbol{K}) - \pi(\boldsymbol{t}_j^*, \boldsymbol{K})) - (\pi(\hat{\boldsymbol{t}}_i, \hat{\boldsymbol{K}}) - \pi(\hat{\boldsymbol{t}}_j, \hat{\boldsymbol{K}})) \right\|, \tag{11}$$

where the function $\pi(\boldsymbol{x}, \boldsymbol{K})$ projects a 3D point $\boldsymbol{x}$ to 2D image coordinates using a camera matrix $\boldsymbol{K}$ described in Section A.2. The first difference in Equation 11 measures velocity for object with corrected translation, while the second difference measures the velocity for object with ground-truth

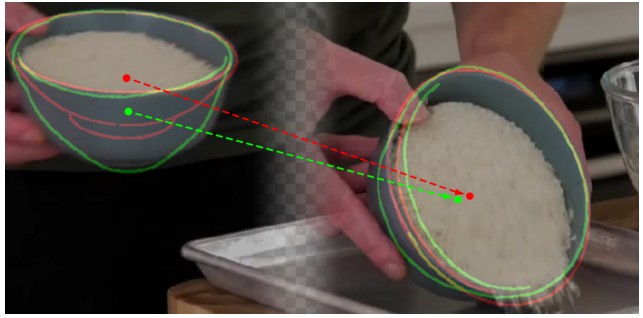

Figure 7: **Visualization of object center projection.** When evaluating translation and depth error, the ground-truth (green) and estimated (red) meshes can have different object centers, causing the spatial velocity between two frames to differ. We illustrate this in the image by showing a manipulation of an object between two different frames $i$ (left) and $j$ (right) with visualization of trajectories of two different meshes representing the motion of the same manipulated object. We overlay the video frames with mesh contours and projected object centers, and corresponding velocities (dashed lines). Even though both red and green objects reasonably describe the motion of the manipulated object, the spatial velocities of the projected objects are different, requiring a change of the object coordinate frame to prevent an unintended increase of the measured errors.

Table 5: **Comparison with the state-of-the-art in 6D object tracking in the wild on the depth metric.** The 6D object tracking results of our approach are compared to MegaPose (Labbé et al., 2022) and FoundPose (Örnek et al., 2024). Even though our method outperforms baselines on the depth metric, note that in our experiments, we did not find the depth error to be informative. In particular, all compared methods can be outperformed by a simple baseline that has zero velocity in the depth direction, resulting in depth error of 0.0289. We hypothesize this could be due to, e.g. the manipulated objects in both instructional and egocentric videos tend to move less in the depth direction, and due to difficulty of the manual ground-truth object annotation performed by visually aligning an approximate object mesh in the video.

| | MegaPose (coarse) | MegaPose | FoundPose | Ours | Ours (refined) |
|---|---|---|---|---|---|
| Relative depth ↓ | 0.1198 | 0.1192 | 0.2592 | 0.0801 | 0.0800 |

translation. To consistently compare methods on videos of different resolutions, we normalize the error and express it as a percentage of the video's diagonal size.

**Depth error.** In a similar manner as the other two metrics, we measure spatial velocity error of the estimated and ground-truth depths. However, normalized by the object scales $s$ and $\hat{s}$:

$$e_{t_1,t_2}^{\text{depth}} = \left\| (d_i - d_j)/s - (\hat{d}_i - \hat{d}_j)/\hat{s} \right\|, \tag{12}$$

where $d_i = \left\| \boldsymbol{t}_i^* \right\|$ is the estimated depth and $\hat{d}_i = \left\| \hat{\boldsymbol{t}}_i \right\|$ is the ground-truth depth. In our experiments, we have found this metric to be insufficiently informative, as all compared methods can be easily outperformed by a trivial assumption that the manipulated object does not move in the depth direction at all, achieving a depth error of 0.0289 compared to 0.0801 of our method. We hypothesize that this is due to two factors. First, objects in both instructional and egocentric videos tend to move less in the depth direction. Second, it is hard for a human annotator to annotate depth by visually aligning an object in the video, especially when the aligned object does not perfectly match the shape of the depicted object. We show the complementary depth metric results in the Table 5.

Table 6: **Influence of different 3D model retrieval methods on 6D pose estimation.** Oracle (gray) shows an upper bound on the performance when the exact meshes of the objects depicted in the image and their scales are known. Our approach outperforms the multimodal OpenShape representation. Additionally, we observe that the model achieves the strongest performance using `FFA` features from layer 22 of the DINOv2 model.

| Retrieval Method | AR↑ | AR_CoU↑ | AR_CH↑ | AR_pCH↑ |
|---|---|---|---|---|
| OpenShape (Liu et al., 2024a) | 34.51 | 20.25 | 10.59 | 72.69 |
| Ours (layer 18 - `CLS`) | 46.42 | 39.78 | 12.49 | 86.99 |
| Ours (layer 20 - `CLS`) | 48.36 | 43.46 | 13.97 | **87.66** |
| Ours (layer 22 - `CLS`) | 45.05 | 40.93 | 16.16 | 78.06 |
| Ours (layer 24 - `CLS`) | 41.14 | 36.10 | 15.62 | 71.68 |
| Ours (layer 18 - `FFA`) | 47.40 | 43.68 | 15.35 | 83.16 |
| Ours (layer 20 - `FFA`) | 48.46 | 44.67 | 17.37 | 83.33 |
| Ours (layer 22 - `FFA`) | **49.86** | **45.20** | **18.53** | 85.83 |
| Ours (layer 24 - `FFA`) | 46.73 | 42.04 | 17.32 | 80.84 |
| Oracle - `CLS` | 63.31 | 52.42 | 46.43 | 91.07 |
| Oracle - `FFA` | 62.93 | 51.99 | 45.95 | 90.85 |

## C  ADDITIONAL ABLATIONS AND QUANTITATIVE RESULTS

### C.1  ABLATION OF CAD MODEL RETRIEVAL AND ALIGNMENT

In Table 6 we explore how the overall performance of the pipeline changes by choosing tokens from different layers of the DINOv2 model. We also provide comparison to using `CLS` token baseline inspired by CNOS (Nguyen et al., 2023). In our experiments, we observe that doing retrieval and pose estimation using patch features from the layer 22 of the DINOv2 model yields the best performance. Compared to the related work of FoundPose (Örnek et al., 2024), which used layer 18 in their pipeline, our method achieves optimal performance using layer 22 features, indicating that our task needs more semantic information, which is predominately in the last layers of ViT models (Amir et al., 2022).

### C.2  EFFECT OF PROPOSAL GENERATION ON SCALE ESTIMATION

Our scale estimation approach uses all detected objects in the image to compute the absolute object sizes. If the detections are of poor quality, the estimated object sizes will be incorrect. Therefore, we validate object proposal methods on the YCB-V dataset using the standard object detection metrics (average precision and recall) as well as the median scale error relative to the ground-truth. In Table 7, we show that our approach based on an off-the-shelf object detector and mask segmenter significantly outperforms the widely used proposal generator method CNOS (Nguyen et al., 2023). Crucially, we observe the CNOS method oversegments the scene and produces a large amount of spurious proposals. The number of spurious proposals of CNOS can be over 50%, negatively affecting the object scale estimation even as our approach uses an outlier rejection method to cope with incorrect proposals and scale estimates. Finally, we also show the scale estimation error when using the ground truth object masks. Interestingly, the scale estimation error in this case is worse than using our method to obtain the masks. This can be explained by the fact our method detects additional objects in the background, such as chairs, that are not in the ground truth masks but help with the scale estimation.

### C.3  ABLATION OF NUMBER OF VIEWPOINTS

To examine the effect of the rotation sampling on the overall pipeline performance, we perform an ablation study on YCB-V dataset shown in Table 8 in which we increase the number of sampled views in the utilized sampling strategy (Alexa, 2022). In our method, we use $N = 600$ samples, which, on average, leads to a $\sim 25°$ geodesic error between the closest rotations. When increasing

Table 7: **Effect of proposal generation methods on scale estimation, precision, and recall.** We compare our proposal generation approach to the CNOS method (Nguyen et al., 2023) with Fast-SAM segmenter (Zhao et al., 2023).

| Method | Scale error [%] $\downarrow$ | AP $\uparrow$ | AR@10 $\uparrow$ | Proposals per image |
|---|---|---|---|---|
| CNOS (FastSAM) | 118.55 | 61.60 | 71.80 | 34 |
| Ours | **14.42** | **69.70** | **76.70** | 6 |
| GT Masks | 16.65 | 100 | 100 | - |

$N$ to $N = 1200$ or $N = 1800$, we observe that the overall pipeline performance remains similar, while the computational requirements for storage and runtime increase linearly. Our prerendered mesh database takes $\sim$1TB of disk space for $N = 600$ views, $\sim$2 TB for $N = 1200$ views and $\sim$3 TB for $N = 1800$ views. The runtime also scales linearly from $\sim$0.2s per object to $\sim$0.4s per object for 1200 views and $\sim$0.6s per object for 1800 views.

Table 8: **Ablation of Number of Viewpoints.** The overall pose estimation performance compared to our choice of $N = 600$ views is not substantially affected by increasing the number of rendered viewpoints of each object, while the overall runtime and storage requirements increase linearly.

| $N_{samples}$ | AR $\uparrow$ | $AR_{CoU} \uparrow$ | $AR_{CH} \uparrow$ | $AR_{pCH} \uparrow$ | Avg Err |
|---|---|---|---|---|---|
| 600 | 49.86 | 45.20 | 18.53 | 85.83 | $\sim$25° |
| 1200 | 49.66 | 44.64 | 19.10 | 85.24 | $\sim$20° |
| 1800 | 49.10 | 43.95 | 18.25 | 85.11 | $\sim$16° |

## C.4 LLM Model Ablation

To evaluate the dependence of our scale estimation method on the choice of LLM, we conducted an additional ablation study using three other models: one proprietary (GPT-3.5-Turbo) and two open-source models (Llama-3.1-8B Llama Team (2024) and Gemma2-9B Gemma Team (2024)). We adapted prompts for each model to ensure the generation of a scale database in the expected format and the best possible coverage of real-world objects.

The results of our experiments are shown in the Table 9. We observe that GPT-4 outperforms all other models, with Llama-3.1-8B achieving a similar score. Surprisingly, GPT-3.5-Turbo performs the worst, while the Gemma2-9B model is slightly better than GPT-3.5-Turbo but still lags behind GPT-4 and Llama-3.1-8B. Please note that scale estimation does not affect the projected metrics and therefore $AR_{CoU}$ and $AR_{pCH}$ remain constant.

Table 9: **Ablation of LLM Model.** While GPT-4 performs the best, the open-source Llama-3.1-8B model achieves a similar performance. The GPT-3.5-Turbo and Gemma2-9B models perform significantly worse in our study.

| LLM model | AR $\uparrow$ | $AR_{CoU} \uparrow$ | $AR_{CH} \uparrow$ | $AR_{pCH} \uparrow$ |
|---|---|---|---|---|
| GPT-4 | 49.86 | 45.20 | 18.53 | 85.83 |
| GPT-3.5-Turbo | 45.29 | 45.20 | 4.85 | 85.83 |
| Llama-3.1-8B | 49.32 | 45.20 | 16.92 | 85.83 |
| Gemma2-9B | 46.00 | 45.20 | 6.98 | 85.83 |

## C.5 Results on DTTD2 Dataset

In Table 10 we provide additional quantitative results on the DTTD2 (Huang et al., 2023) dataset, which is a dataset featuring 100 scenes and 18 objects, which partially overlap with the widely used YCB-V dataset. On this dataset, our method outperforms all previous state-of-the-art methods by a large margin.

Table 10: **Comparison with the state-of-the-art on the DTTD2 dataset.** The 6D pose estimation results of our approach are compared to MegaPose (Labbé et al., 2022), GigaPose (Nguyen et al., 2024), and FoundPose (Örnek et al., 2024). Our method outperforms all other methods in all the metrics.

| Method | AR↑ | $AR_{CoU}$↑ | $AR_{CH}$↑ | $AR_{pCH}$↑ |
|---|---|---|---|---|
| MegaPose (coarse) | 26.15 | 13.72 | 6.67 | 58.05 |
| MegaPose | 27.84 | 17.73 | 8.77 | 57.00 |
| GigaPose | 41.84 | 39.32 | 12.16 | 74.00 |
| FoundPose | 37.39 | 31.32 | 11.41 | 69.44 |
| Ours | **47.11** | **52.10** | **12.68** | **76.54** |

# D    ADDITIONAL QUALITATIVE RESULTS

In Figures 8 and 9, we provide more qualitative results on the EPIC-Kitchens dataset, showing that our approach can work from an egocentric point of view. In Figures 10 and 11, we provide additional qualitative results featuring extraction of object poses from the Internet videos and their transfer to the real robot. The robot trajectory is always computed to closely imitate the relative 6D pose transformations of the detected object from the video. We only transform the trajectory from the camera frame into the robot frame to account for differences in scale and robot position. The perceived differences between the estimated object trajectory from the video, the object trajectory executed by the robot in the simulation and the object trajectory executed by the real robot are due to the differences of camera viewpoints.

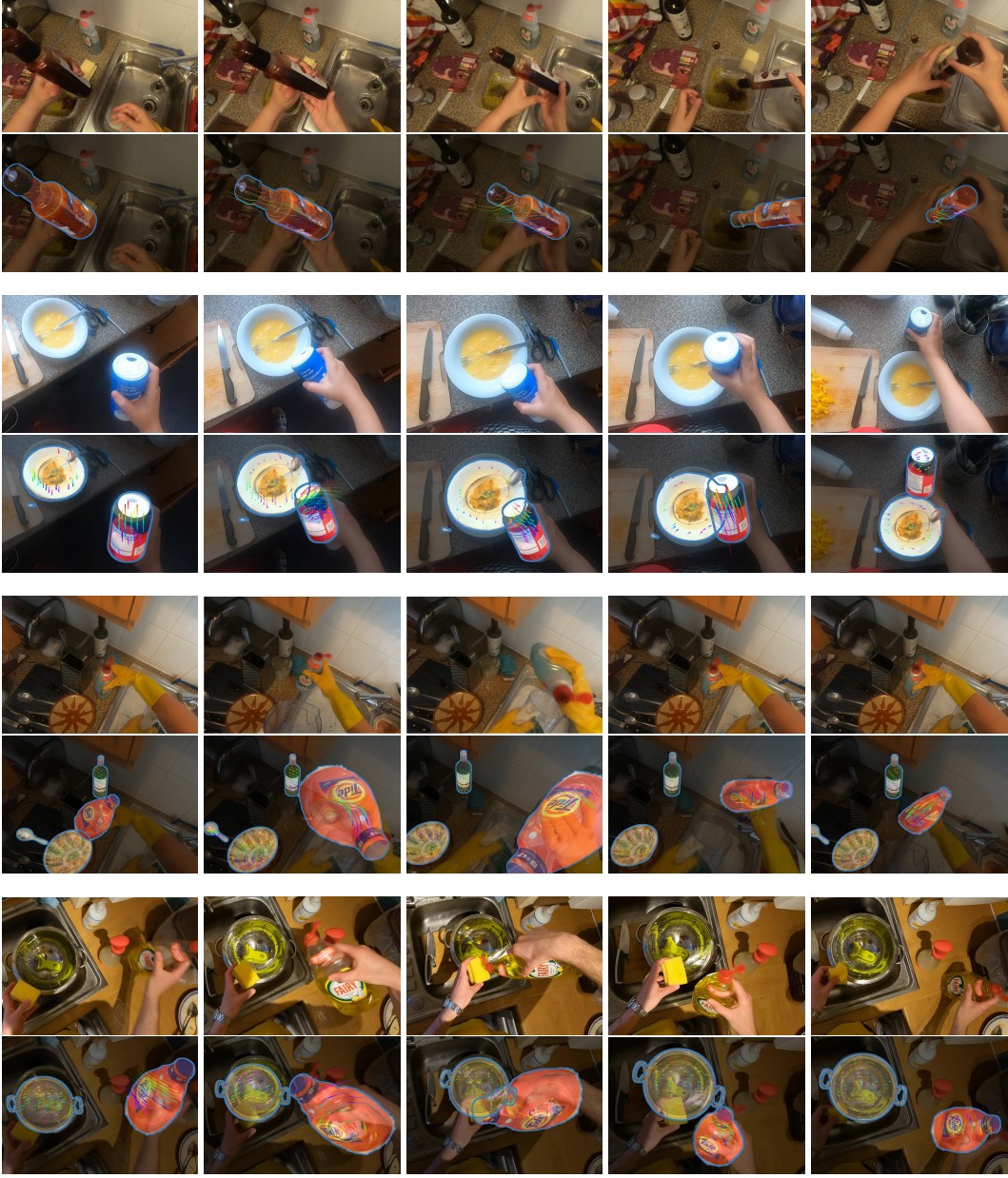

Figure 8: **Additional results of our method on the EPIC-Kitchens dataset.** Our method is able to detect and align objects from an egocentric point of view. We show frames from the input video in the top row of each example, and the tracked 6D poses of the retrieved object together with 2D point tracks in the second row of each example. Please note the differences in object geometry and texture between the objects depicted in the video and the retrieved objects from the database.

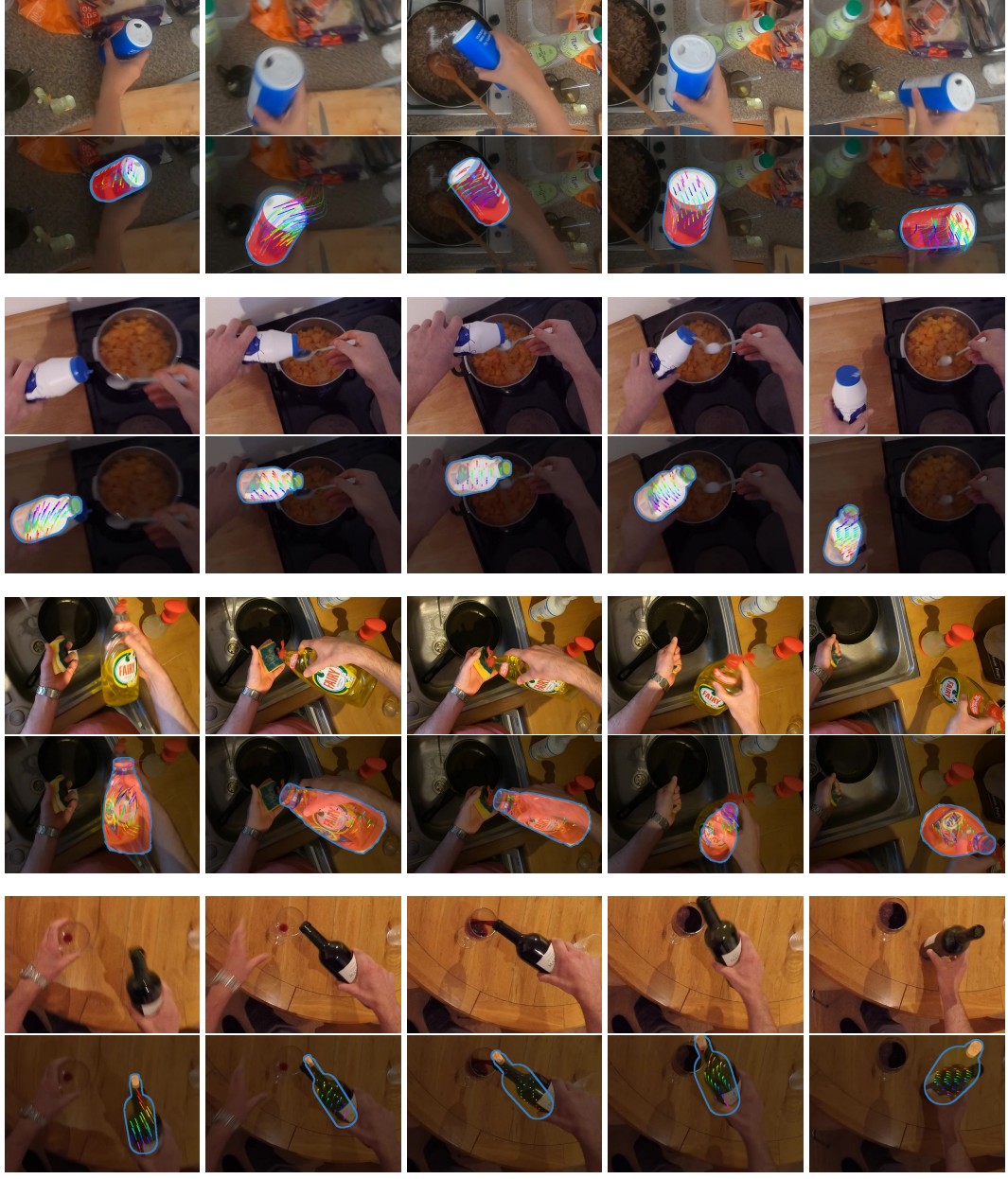

Figure 9: **Additional results of our method on the EPIC-Kitchens dataset.** Our method is able to detect and align objects from an egocentric point of view. We show frames from the input video in the top row of each example, and the tracked 6D poses of the retrieved object together with 2D point tracks in the second row of each example. Please note the differences in object geometry and texture between the objects depicted in the video and the retrieved objects from the database.

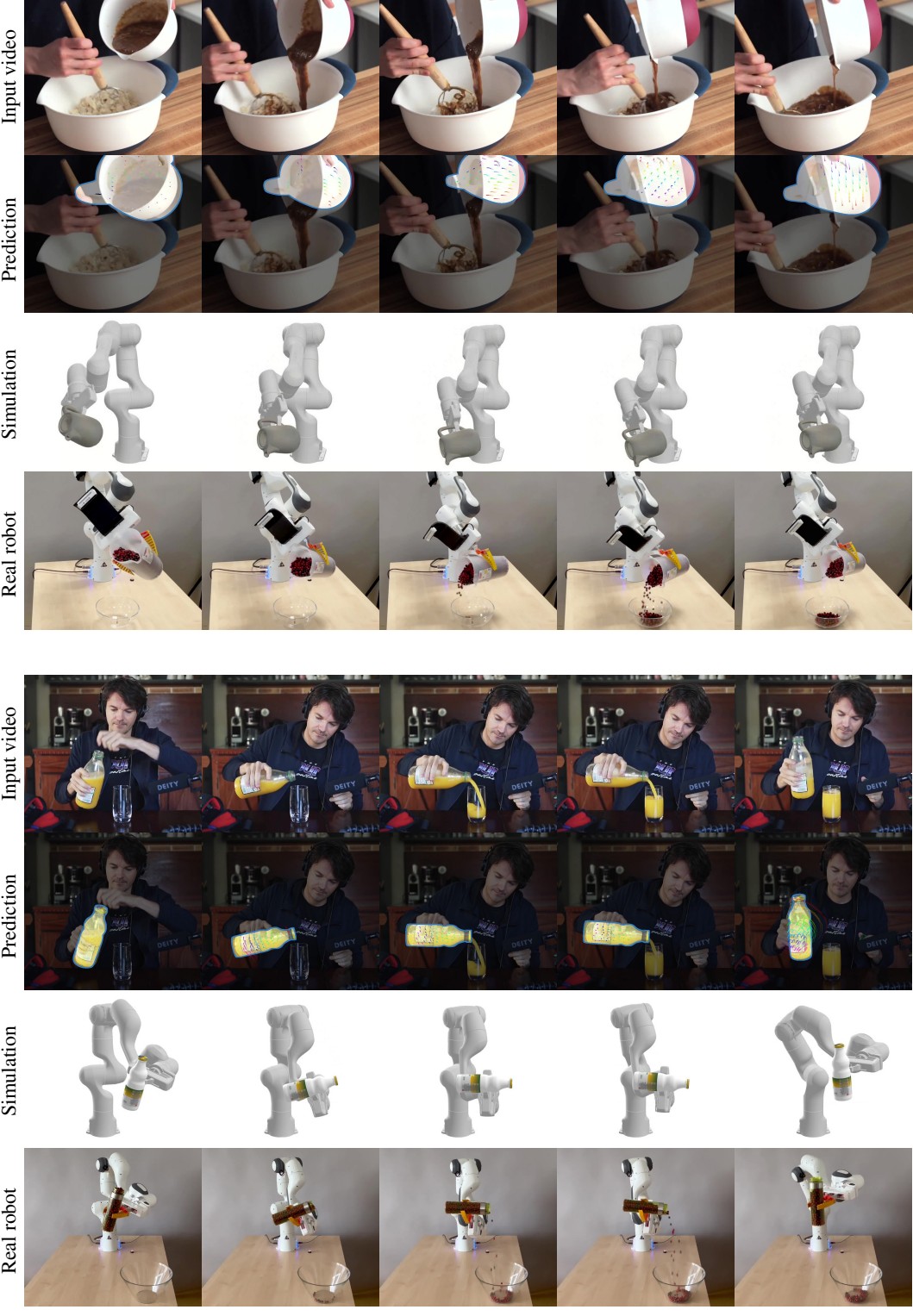

Figure 10: **Additional qualitative results on Internet videos.** For each video, we first show several input frames, followed by the alignment obtained by our method together with tracked 2D-3D correspondences shown as colored tracks. Please note the differences in object geometry and texture between the objects depicted in the video and the retrieved (and aligned) objects from the database. Temporally smooth poses are used to compute robot motion via trajectory optimization as shown in the third row, followed by the video of a real robot execution.

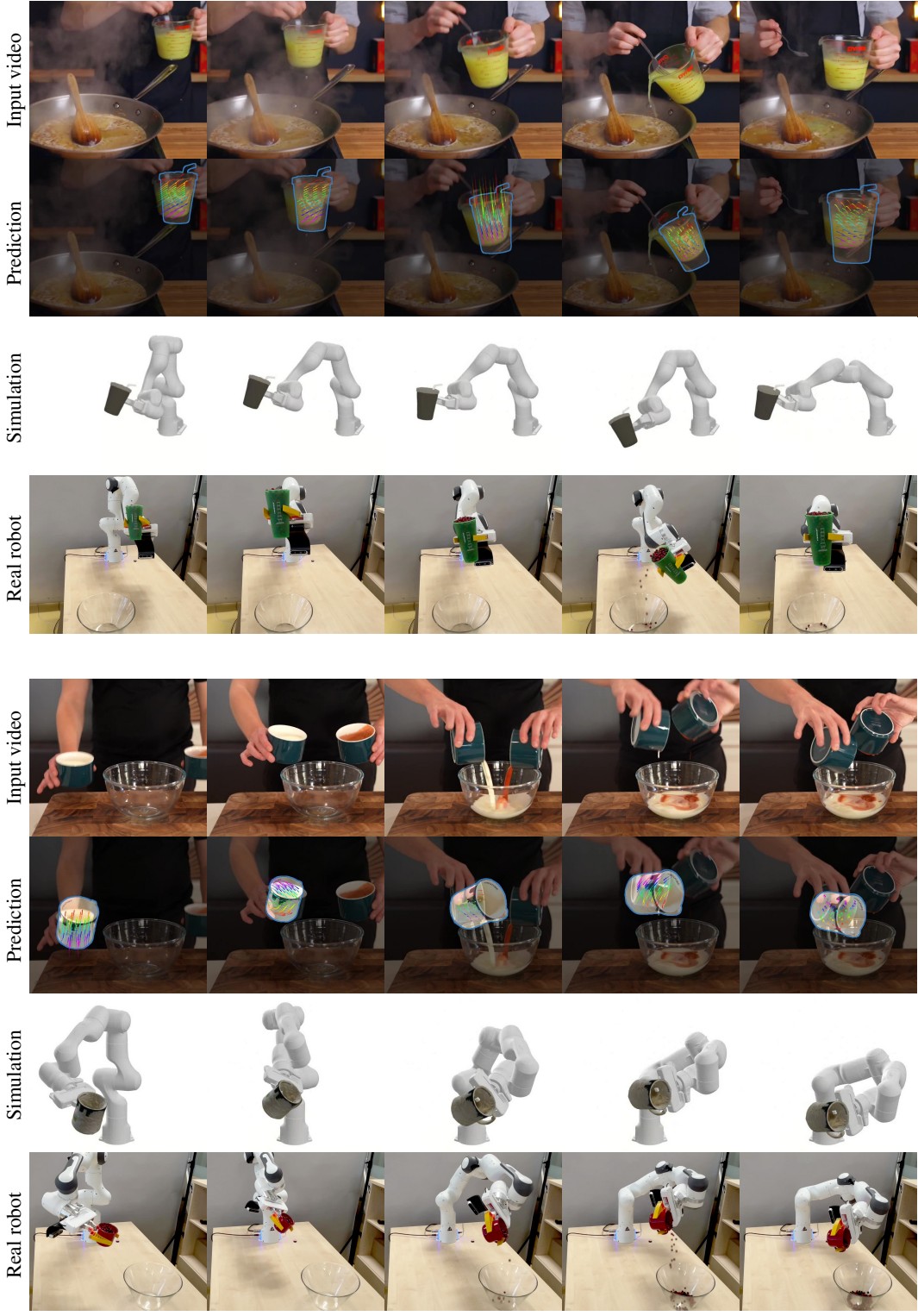

Figure 11: **Additional qualitative results on Internet videos.** For each video, we first show several input frames, followed by the alignment obtained by our method together with tracked 2D-3D correspondences shown as colored tracks. Please note the differences in object geometry and texture between the objects depicted in the video and the retrieved (and aligned) objects from the database. Temporally smooth poses are used to compute robot motion via trajectory optimization as shown in the third row, followed by the video of a real robot execution.

