# OpenReview forum: "6D Object Pose Tracking in Internet Videos for Robotic Manipulation"
_ICLR.cc/2025/Conference — ICLR 2025 Poster_

### Official Review · Reviewer_TaH5 · 2024-10-31

**Soundness:** 3
**Presentation:** 3
**Contribution:** 3
**Rating:** 6
**Confidence:** 4

**Summary:**

The paper pays attention on 6D pose trajectory estimation of a manipulated object from an Internet instructional video with a novel framework. The framework first predicts the 6D pose of any object by CAD model retrieval. Then the smooth 6D object trajectories are extracted and retargeted via trajectory optimization into a robotic manipulator. Experiments on YCB-V and HOPE-Video datasets demonstrate the improvements over RGB 6D pose methods. Moreover, the 6D object motion can be transferred to a 7-axis robotic manipulator.

**Strengths:**

1 The pose estimation method by retrieving a CAD model, aligning the retrieved CAD model with the object, and grounding the object scale with respect to the scene.

2 Consistent 6D pose trajectory estimation from Internet videos and retargeting trajectories to a robotic manipulator.

3 The pose estimation improvement on YCB-V and HOPEVideo datasets, and transfer from 6D object motion to a 7-axis robotic manipulator.

**Weaknesses:**

1 The original contributions should be expressed more clearly. In the proposed method, various existing methods are employed. It is suggested to clearly distinguish the original contributions in this paper and usage of other methods. Specifically, the first contribution locates in the pose estimation method by retrieving a CAD model, aligning the retrieved CAD model, and grounding the object scale with respect to the scene. The subsequent question is that what is the original contribution, the whole pipeline or the detailed design of a particular module? The authors are suggested to express this more clearly in the revised version. For the second and third contributions, it is also recommended to present more clear expressions.

2 For robotic manipulation, the running time of the pose estimation method is a key factor. The proposed method in the paper is somewhat time-consuming with 2s for detector, retrieval and scale estimation per scene and 0.2s for pose estimation per object. To further improve the paper, two suggestions are given. For one thing, the comparaions with other methods on running time are suggested to add.  For another, more analysis about the running time is also preferred, such as the recommendations for accelerate the whole method.

**Questions:**

1 With the similar CAD model retrieval, the classification can also be obtained. I wonder if it is possible to use the CAD model to perform classification directly?

---

> ### Author Response · Authors · 2024-11-21
>
> ### **Weaknesses**
>
> > The original contributions should be expressed more clearly. In the proposed method, various existing methods are employed. It is suggested to clearly distinguish the original contributions in this paper and usage of other methods. Specifically, the first contribution locates in the pose estimation method by retrieving a CAD model, aligning the retrieved CAD model, and grounding the object scale with respect to the scene. The subsequent question is that what is the original contribution, the whole pipeline or the detailed design of a particular module? The authors are suggested to express this more clearly in the revised version. For the second and third contributions, it is also recommended to present more clear expressions.
>
> **Response:**
> Thank you for letting us address this concern. The first and main contribution of our the paper lies in designing an end-to-end approach that addresses in-the-wild pose estimation in a completely novel setup without the available exact mesh. While we build on existing work, we have developed an approach that significantly outperforms other methods on this challenging problem. We find it interesting that the existing best 6D pose estimation methods can be significantly outperformed on this problem by our simple yet carefully designed 2D-3D retrieval/matching approach combined with robust scale estimation and 2D-3D tracking.
>
> Second, addressing this hard problem required the development of a new module for object scale estimation, a challenge that has not been adequately explored in the literature, as well as adaptations to the object detection stage. Lastly, we investigated and introduced metrics suitable for comparing different methods, as traditional 6D pose estimation metrics are not applicable in this setup without known exact mesh. We will clarify the novelty of our approach more explicitly in the revised version.
>
> > For robotic manipulation, the running time of the pose estimation method is a key factor. The proposed method in the paper is somewhat time-consuming with 2s for detector, retrieval and scale estimation per scene and 0.2s for pose estimation per object. To further improve the paper, two suggestions are given. For one thing, the comparaions with other methods on running time are suggested to add. For another, more analysis about the running time is also preferred, such as the recommendations for accelerate the whole method.
>
> **Response:**
> Our method is primarily an offline method that is meant for offline extraction of manipulation trajectories from videos for (offline) robot learning. Moreover, the detector is used only on the first frame of the video, and the objects are tracked through the video with \[SAM2\], which can run in real time. Then we run the retrieval and scale estimation, which can benefit from multi-frame prediction aggregation, but does not necessarily have to use all video frames, especially for longer videos. In practice, we used 30 frames throughout the video for the retrieval and scale estimation.
>
> Our method is being run in BF16 to speed up the inference time. Given that our method processes every object independently, the inference can be parallelized among multiple processes and GPUs.
>
> For runtime comparison, on average, for a single video frame our method takes \~0.2s per object, while MegaPose takes \~4.5s per object, GigaPose takes 0.03s per object and FoundPose around \~0.15s per object. We will include runtimes of all methods used in the paper into the revised manuscript.
>
> \[SAM2\] Ravi, Nikhila, et al. "Sam 2: Segment anything in images and videos." arXiv preprint arXiv:2408.00714 (2024).
>
> ### **Questions**
>
> > With the similar CAD model retrieval, the classification can also be obtained. I wonder if it is possible to use the CAD model to perform classification directly?
>
> **Response:**
> Indeed, there are multiple ways to tackle the CAD model retrieval. In our paper, we explored retrieval based on averaged visual features (\[CLS\] tokens of the DINOv2 model and FFA descriptors constructed from foreground patch tokens of the DINOv2 model). We also add a multi-modal OpenShape model for comparison, which aims to align image-text-mesh triplets in CLIP feature space. The retrieval using this model is done by matching the CLIP image token to the \~50,000 precomputed CAD models projected into CLIP space using OpenShape, which can be understood as zero-shot classification. However, in Table 2 we show that for our task, the image-based retrieval outperforms the OpenShape baseline. We assume that this is caused by the fact that OpenShape is trained with synthetic renderings, which causes a large domain gap between the training data and real in-the-wild images of objects.

---

> > ### Comment · Reviewer_TaH5 · 2024-11-26
> >
> > Thank you for your detailed response. The response has successfully addressed most of my concerns. By combining my opinion with the comments of other reviewers, I decide to keep my original score 6.

---

### Official Review · Reviewer_4DS1 · 2024-11-03

**Soundness:** 3
**Presentation:** 3
**Contribution:** 3
**Rating:** 6
**Confidence:** 4

**Summary:**

The authors present a novel approach to extract temporally consistent 6D pose trajectories of manipulated objects from Internet videos to be applied with robotic manipulation task. It tackles the challenges posed by uncontrolled capture conditions, unknown object meshes, and complex object motions. Their evaluation on YCB-V and HOPE-Video datasets shows state-of-the-art performance, with successful motion transfer to a robotic manipulator in both simulated and real-world settings.

----------------------------------------------------------------------------------------------------
The authors addressed most of my concerns in the rebuttal phase, and thus I would like to raise my score to 6.

**Strengths:**

The impact of the paper is dominant in the way that it provides an envision of enriched data for robotic manipulation without human labor force to construct the specific datasets. The methodology is intuitive and the performance enhancement is non-trivial. The paper is overall well-written.

**Weaknesses:**

My primary concern lies with the methodological novelty, as the approach largely involves applying an existing pipeline to internet videos. Specifically, the use of an LLM for estimating object scale may be questionable, given potential uncertainties around its accuracy in providing a realistic scale for each object. Aside from this, the methodology essentially adapts previous methods to fit the proposed pipeline. Given these factors, I feel this work might not align with ICLR's focus but could be more suited to a robotics conference.

**Questions:**

1. It might be great if the authors could ablate on the performance variation under different LLMs. Currently it only applies GPT-4, but it is important to know how different LLMs might influence the performance (i.e. one GPT-3.5 & one open-source LLM).
2. What's the efficiency & cost of such pipeline when performing inference on a 1-minute Instructional videos?
3. Using a CAD model can be costly since it requires a large database to store predefined meshes, and in open-world scenarios, finding an exact match is often unlikely. However, numerous approaches avoid relying on CAD models. For instance, "6DGS: 6D Pose Estimation from a Single Image and a 3D Gaussian Splatting Model" [ECCV 2024]. Have you tried experimenting with such methods? Or say, how do you envision those methods' strengths and weaknesses compared to your method.
4. For the standard evaluation, it might be beneficial to add another dataset evaluation using different cameras, say iPhone sensor as proposed in "Robust 6DoF Pose Estimation Against Depth Noise and a Comprehensive Evaluation on a Mobile Dataset" to further validate the approach's generalizability.

---

> ### Author Response · Authors · 2024-11-21
>
> ### **Weaknesses**
>
> > My primary concern lies with the methodological novelty, as the approach largely involves applying an existing pipeline to internet videos. Specifically, the use of an LLM for estimating object scale may be questionable, given potential uncertainties around its accuracy in providing a realistic scale for each object. Aside from this, the methodology essentially adapts previous methods to fit the proposed pipeline. Given these factors, I feel this work might not align with ICLR's focus but could be more suited to a robotics conference.
>
> **Response:**
> Thank you for letting us clarify this concern. Our method focuses on a novel in-the-wild setup of 6D pose estimation, that estimates the pose of the depicted object without prior knowledge of the object itself. While we build on existing work, we have developed an approach that significantly outperforms other methods on this challenging problem. We find it interesting that the existing best 6D pose estimation methods can be significantly outperformed on this problem by our simple yet carefully designed 2D-3D retrieval/matching approach combined with robust scale estimation and 2D-3D tracking.   In addition, the issue of scale estimation has not been adequately addressed in the literature, and LLMs provide a viable path to capture prior knowledge about the visual world. Our approach allows us to robustly map that prior knowledge to specific objects in the specific observed scene while carefully addressing the inherent noise and ambiguities in the problem. Although scale estimates for individual objects are noisy, our method aims to robustly mitigate these object-level errors by aggregating information from multiple objects in the entire scene, as explained in the “Global Rescaling” paragraph of Section A.3.
>
> ### **Questions**
>
> > It might be great if the authors could ablate on the performance variation under different LLMs. Currently it only applies GPT-4, but it is important to know how different LLMs might influence the performance (i.e. one GPT-3.5 & one open-source LLM).
>
> **Response:**
> To evaluate the dependence of our scale estimation method on the choice of LLM, we conducted an additional ablation study using three other models: one proprietary (GPT-3.5-Turbo) and two open-source models (Llama-3.1-8B and Gemma2-9B). We adapted prompts for each model to ensure the generation of a scale database in the expected format and the coverage of real-world objects.
>
> The results of our experiments are shown below. We observe that GPT-4 outperforms all other models, with Llama-3.1-8B achieving a very similar score. Surprisingly, GPT-3.5-Turbo performs the worst, while the Gemma2-9B model is slightly better than GPT-3.5-Turbo but still lags behind GPT-4 and Llama-3.1-8B. Please note that scale estimation do not affect the projected metrics and therefore $AR_{CoU}$  and  $AR_{pCH}$ remain constant.
>
> | LLM model | AR | AR$_{CoU}$ | AR$_{CH}$ | AR$_{pCH}$ |
> | :---- | :---- | :---- | :---- | :---- |
> | GPT-4 | 49.86 | 45.20 | 18.53 | 85.83 |
> | GPT-3.5-Turbo | 45.29 | 45.20 | 4.85 | 85.83 |
> | Llama-3.1-8B | 49.32 | 45.20 | 16.92 | 85.83 |
> | Gemma2-9B | 46.00 | 45.20 | 6.98 | 85.83 |
>
> > What's the efficiency & cost of such pipeline when performing inference on a 1-minute Instructional videos?
>
> **Response:**
> The method onboarding stage consists of pre-rendering the meshes and extracting the features. When running on a single image, the resulting runtime for detection, retrieval, and scale estimation is \~2s per image. For the pose estimation part, the runtime is \~0.2s per object. However, when running on an instructional video, we run the detector only on the first frame and track the detected objects through the video using \[SAM2\], which can run in real-time. Then we run the retrieval and scale estimation, which can benefit from multi-frame prediction aggregation, but does not necessarily have to use all video frames, especially for longer videos. In practice, we used 30 frames throughout the video for the retrieval and scale estimation. In contrast to single-image estimation, we use \[CoTracker\] combined with PnP to extract smooth poses from the video, with run-time of \~1s per frame.
>
> \[SAM2\] Ravi, Nikhila, et al. "Sam 2: Segment anything in images and videos." arXiv preprint arXiv:2408.00714 (2024).
> \[CoTracker\] Karaev, Nikita, et al. ‘CoTracker: It Is Better to Track Together’. Proc. ECCV, 2024\.

---

> ### Author Response · Authors · 2024-11-21
>
> > Using a CAD model can be costly since it requires a large database to store predefined meshes, and in open-world scenarios, finding an exact match is often unlikely. However, numerous approaches avoid relying on CAD models. For instance, "6DGS: 6D Pose Estimation from a Single Image and a 3D Gaussian Splatting Model" \[ECCV 2024\]. Have you tried experimenting with such methods? Or say, how do you envision those methods' strengths and weaknesses compared to your method.
>
> **Response:**
> Thank you for the question. Even though 6DGS does not need a CAD model of the object, it requires a 3DGS model, i.e. gaussian splatting representation of the object/scene. However, creating the 3DGS model requires having a set of images of a static scene from calibrated cameras with the corresponding camera poses. When running on in-the-wild images or YouTube videos, we do not have this information and thus cannot create the GS model in the same way. In contrast, our method estimates the poses/trajectories of objects in-the-wild, and does not require prior information about the object. We agree that the exact CAD model is often not available even in large-scale object databases like Objaverse, however, our method leverages the power of DINOv2 features to estimate the poses of observed objects even with an approximate CAD model. The intuition behind our approach is that we can estimate the *relative* 6D object transformations from the video (i.e. the object’s trajectory) even if the matched object from the database is not exact.
>
> At the initial stage of our project we explored potential applications of generation image-to-3D models such as \[zero123-XL\] or \[stable-zero123\], we found that in practice the CAD models generated by those models conditioned on in-the-wild real-world photos are of insufficient quality for our task. This is mainly driven by the fact that lots of query images are taken from varying elevation angles, lightning conditions etc, which makes inconsistent predictions with those models.
>
> We have also experimented with methods such as \[Dust3r\] or \[SplatterImage\] to build 3D object models directly from the input Youtube videos and found that those methods did not produce usable results (often producing severely distorted objects) likely due to the challenging nature of such in-the-wild videos (unusual viewpoints, difficult illumination, occlusions, low resolution, blur, etc).
>
> \[zero123-XL\] Ruoshi Liu et al., Zero-1-to-3: Zero-shot One Image to 3D Object, ICCV 2023
> \[stable-zero123\] Stability AI,. Stable-Zero123
> \[Dust3r\] DUSt3R: Geometric 3D Vision Made Easy, Shuzhe Wang, Vincent Leroy, Yohann Cabon, Boris Chidlovskii, Jerome Revaud, CVPR 2024
> \[SplatterImage\] Splatter image: Ultra-fast single-view 3d reconstruction, Stanislaw Szymanowicz, Chrisitian Rupprecht, Andrea Vedaldi, CVPR 2024\.
>
> > For the standard evaluation, it might be beneficial to add another dataset evaluation using different cameras, say iPhone sensor as proposed in "Robust 6DoF Pose Estimation Against Depth Noise and a Comprehensive Evaluation on a Mobile Dataset" to further validate the approach's generalizability.
>
> **Response:**
> Thank you for suggesting this dataset for evaluating our approach. We converted the DTTD2 dataset into the standardized BOP format and evaluated it using our setup, which uses an RGB-based model-retrieved pose estimation. The results of our method, along with all baseline methods, are presented in the table below. Please note that FoundPose results were obtained using the official source code by the authors of the method. This is in contrast to the submission, where we used our own implementation because the official code was not yet available at the time of the submission.
>
> The observed trend on this new dataset is consistent with the other datasets we evaluated on: our method achieves the highest recall, as baseline methods struggle with inaccuracies in the mesh compared to our approach. Note that we used our model retrieval and scale estimation modules to generate the same scaled mesh for all methods in the table.
>
> | Method | $AR$ | $AR\_{CoU}$ | $AR\_{CH}$ | $AR\_{pCH}$ |
> | :---- | :---- | :---- | :---- | :---- |
> | MegaPose (w/o refiner) | 26.15 | 13.72 | 6.67 | 58.05 |
> | MegaPose  | 27.84 | 17.73 | 8.77 | 57.00 |
> | GigaPose | 41.84 | 39.32 | 12.16 | 74.00 |
> | FoundPose | 37.39 | 31.32 | 11.41 | 69.44 |
> | Ours | **47.11** | **52.10** | **12.68** | **76.54** |

---

> > ### Comment · Reviewer_4DS1 · 2024-11-25
> >
> > The authors addressed most of my concerns in the rebuttal phase, and thus I would like to raise my score to 6.

---

### Official Review · Reviewer_JJFi · 2024-11-03

**Soundness:** 3
**Presentation:** 3
**Contribution:** 3
**Rating:** 6
**Confidence:** 4

**Summary:**

This paper proposes a new approach to detect and track the 6-DoF pose of unknown objects from RGB video. The approach is motivated by robot imitation learning from internet video. The approach uses off-the-shelf open-set object detectors, foundation models for segmentation, vision-language (CLIP), and visual features (DINOv2) to detect objects, retrieve similar shapes from a database of CAD models, and matching the object image with a set of rendered views of the object CAD model to estimate 3D orientation. Experimental evaluation is performed quantititvely on YCB-Video and HOPE-Video datasets and a comparison is made with state of the art object detectors for unseen objects for which the CAD model is assumed known (MegaPose, GigaPose). Also, qualitative results on EPIC-Kitchen, and an example of executing the estimated object trajectories on a real robot are shown.

**Strengths:**

- The proposed approach for detecting and estimating 6D motion of unknown objects from RGB images is novel and interesting.
- The paper is well written and easy to follow.
- The set of experiments demonstrate the shape retrieval and pose estimation well and also compare with state of the art methods.
- A qualitative example is provided with a real robot which show the robot pouring from one object to another.

**Weaknesses:**

- l. 197ff, CAD model retrieval by rendering views and calculating visual features seems expensive in both, the database generation and the retrieval stage for large datasets such as Objaverse-LVIS. What is the retrieval time for these datasets and how is it implemented to make retrieval efficient?
- l. 220ff proposes to retrieve rotation by matching to a set of rendered views. What is the choice of N in the experiments? What is the avg/std angular distance between sampled rotations?
- l. 243ff, the way to prompt the LLM in the supplementary is an offline procedure to collect size estimates for approximately 2200 objects. In the main paper, the description reads as if the LLM is prompted for each detected object using the CLIP text classification. Please describe this more clearly. What if the detected object is not included in the offline calculated set ?
- l. 286, was estimating the motion of the camera relative to the static background evaluated in this work ? Please clarify.
- The optimization problem in eq 4 does not provide a description of the used system dynamics model.
- l. 361, please write more clearly, that while a similar mesh is known, the retrieved mesh does not exactly correspond to the ground truth mesh which is an assumption used for MegaPose and GigaPose.
- Please introduce the pCH metric formally, at least in the supplemental material. The current description is insufficient.
- l. 519ff, the real robot experiment is rather anecdotal and lacks important details in its descriptions and quantitative evaluation (e.g., success rate). How are the observed object trajectories transfered to the real robot experiment incl. considering the change of view point and embodiment? How does the robot know where the manipulated objects are and how is this matched to the observed object motion?
- Fig. 8, in the upper additional qualitative result, the bowl object pose is not correctly tracked. Why does the robot still turn the object in a quite different angle ?

Additional minor comments:
- Fig. 6, rightmost real robot image seems to be a repetition of the image next to it. Was the wrong image included?

**Questions:**

- l. 323, are the ground-truth meshes contained in the object datasets?
- Table 1, was the same scale estimate for the meshes used for MegaPose and GigaPose like for the proposed method?
- Which dynamics model is used for the optimization problem in eq 4? How is tracking of the optimized trajectory implemented?
- See additional questions in sec. "Weaknesses".

---

> ### Author Response · Authors · 2024-11-21
>
> ### **Weaknesses**
>
> > l. 197ff, CAD model retrieval by rendering views and calculating visual features seems expensive in both, the database generation and the retrieval stage for large datasets such as Objaverse-LVIS. What is the retrieval time for these datasets and how is it implemented to make retrieval efficient?
>
> **Response:**
> While it is true that rendering and extraction of visual features is expensive, this process has to be done only once. The Objaverse-LVIS dataset consists of \~50,000 objects, and the whole dataset can be rendered in \~3 days on one 8-GPU node. Extraction of visual features for retrieval then takes around \~3 days on 8-GPU node as well. In practice, we used the HPC cluster to parallelize and speed up the rendering and extraction process significantly.
>
> The retrieval process is done by matching a single 1024D FFA descriptor extracted from a query image to a database of \~50,000 1024D descriptors (one descriptor for one object) by means of the dot product. This step takes a fraction of a second on a single GPU. The dense features for matching the views are then computed on the fly, and they take around \~0.2s per object on AMD MI250X GPU to extract and match 600 CAD model views to the query image.
>
> > l. 220ff proposes to retrieve rotation by matching to a set of rendered views. What is the choice of N in the experiments? What is the avg/std angular distance between sampled rotations?
>
> **Response:**
> For rotation sampling, we use the strategy described in \[SuperFib\]. We used 600 samples, which resulted in an average geodesic distance error of 25 degrees between the closest rotations, with a standard deviation of 2 degrees. We will include this missing parameter value in the revised paper. We also perform a new ablation study in which we increase the number of samples N to 1200 or 1800\. We observe that the overall performance remains similar, while the computational requirements for storage and runtime increase linearly. Our prerendered mesh database takes \~1TB of disk space for N=600 and \~2 TB for N=1200 views and \~3 TB for N \= 1800 views. The runtime also scales linearly from \~0.2s per object to \~0.4s per object for 1200 views and \~0.6s per object for 1800 views.
>
> | N$_{samples}$ | AR | AR$_{CoU}$ | AR$_{CH}$ | AR$_{pCH}$ | Avg Rot. Err | Std. Dev. Rot. Err |
> | :---- | :---- | :---- | :---- | :---- | :---- | :---- |
> | 600 | 49.86 | 45.20 | 18.53 | 85.83 | \~25 deg | \~2 deg |
> | 1200 | 49.66 | 44.64 | 19.10 | 85.24 | \~20 deg | \~1 deg |
> | 1800 | 49.10 | 43.95 | 18.25 | 85.11 | \~16 deg | \~1.4 deg |
>
> \[SuperFib\] Marc Alexa, Super-Fibonacci Spirals: Fast, Low-Discrepancy Sampling of SO(3), CVPR2022
>
> > l. 243ff, the way to prompt the LLM in the supplementary is an offline procedure to collect size estimates for approximately 2200 objects. In the main paper, the description reads as if the LLM is prompted for each detected object using the CLIP text classification. Please describe this more clearly. What if the detected object is not included in the offline calculated set?
>
> **Response:**
>
> We apologize for the confusion. The LLM prompting happens offline, as we use a fixed set of text labels and corresponding scales for all images and detected objects. After generating the description-scale pairs with the LLM, we pre-extract CLIP features from the text descriptions and store them with their corresponding scales. During inference, we compute CLIP features for the detected objects in the images and retrieve the best-matching text descriptions along with their scales, as detailed in the supplementary material in section A.3.
>
> Importantly, our set of generated text descriptions is not tailored to any specific scene type. Instead, it is designed to encompass a wide range of everyday objects. This set can be easily adapted to specific use cases by generating a new set of descriptions using a modified prompt.
>
> If the generated set lacks text descriptions closely matching the object in the image, the retrieved scales may contain some inaccuracies. However, our method is inherently robust to a certain level of scale error. First, we retrieve multiple matching text descriptions for each detected object and apply median aggregation. Second, even if the aggregated scales are inaccurate (e.g., due to the absence of a closely matching description in the set), these scales are not used directly. Instead, they contribute to computing the global correction factor ρ, which is applied to the relative scales derived from the predicted depth map, as explained in the “Global Rescaling” paragraph of section A.3 in the supplementary material. Thus, when the image contains enough objects covered by the generated set, these objects (with correctly estimated scales) dominate the estimation of the global correction factor, leading to overall improved scale estimates.

---

> ### Author Response · Authors · 2024-11-21
>
> > l. 286, was estimating the motion of the camera relative to the static background evaluated in this work? Please clarify.
>
> **Response:**
> In the paper, we focused on computing the object's motion relative to the camera, so we did not explicitly estimate the camera's relative pose. In the revised version, we will make it clear that camera motion is not estimated. However, the assumption of a static camera is still relevant for many use cases, such as third-person view how-to videos.
>
> > The optimization problem in eq 4 does not provide a description of the used system dynamics model.
>
> **Response:**
> Thank you for bringing this to our attention. We utilized the publicly available dynamic model for the Panda robot from the example-robot-data package, accessible via Conda and PyPI. This package includes the kinematics, as well as the geometric and inertia parameters for each link of the robot.
>
> For forward dynamics computation, we employed the Pinocchio library, which is also internally used within the Aligator trajectory optimization package. The dynamic model was used to compute joint positions and velocities based on the input sequence of joint torques, while the kinematic model was used to determine the object's pose through forward kinematics. Consequently, both the end-effector pose and joint velocities are directly influenced by the optimized torques in Eq. (4). We will explicitly clarify this in the revised version of the paper.
>
> > l. 361, please write more clearly, that while a similar mesh is known, the retrieved mesh does not exactly correspond to the ground truth mesh which is an assumption used for MegaPose and GigaPose.
>
> **Response:**
> Thank you for pointing this out. While we mention this in the related work, we will emphasize this more clearly also in the experiments section in the revised version of the paper.
>
> > Please introduce the pCH metric formally, at least in the supplemental material. The current description is insufficient.
>
> **Response:**
> Thank you for this suggestion. We will expand the definitions of all metrics in Appendix B to include precise formal definitions. The pCH metric is a projected Chamfer distance, inspired by the MSPD (Maximum Symmetry-Aware Projection Distance) metric used in the BOP challenge, given by
>
> $pCH = \\frac{1}{|M_{pred}|} \sum_{x \in M_{pred}}min_{y \in M_{GT}}||\pi(x, K, T) - \pi(y, K, T_{GT})||_2^2$
>
> $~~~~~~~~~+ \\frac{1}{|M_{GT}|} \sum_{y \in M_{GT}}min_{x \in M_{pred}}||\pi(x, K, T) - \pi(y, K, T_{GT})||_2^2$
>
> where $K$ is camera intrinsics, $T$ is a predicted pose, $T_{GT}$ is ground truth pose, $M_{pred}$ is a set of points sampled from the predicted mesh, $M_{GT}$ is a set of points sampled from the ground truth mesh, $x$ and $y$ are vertices of the meshes, and function $\pi$ projects 3D vertex into 2D pixel. The core idea of this metric is to evaluate how well the estimated pose visually aligns when projected onto the image plane. As the closest vertex is found for each vertex of other mesh, this metric can be used for non-identical meshes or for symmetric meshes. This approach allows us to assess alignment quality, even when the predicted scale is not exact, since the scale is factored out during the projection.

---

> ### Author Response · Authors · 2024-11-21
>
> > l. 519ff, the real robot experiment is rather anecdotal and lacks important details in its descriptions and quantitative evaluation (e.g., success rate). How are the observed object trajectories transfered to the real robot experiment incl. considering the change of view point and embodiment? How does the robot know where the manipulated objects are and how is this matched to the observed object motion?
>
> **Response:**
> We apologize for not describing the real-robot experiment in detail. We will address this in the revised version based on the information provided below.
>
> **Moving the object to the starting pose.** In the first step, we manually placed the object into the gripper (fixing the robot gripper-to-object transformation) and moved the object using the robot to the initial pose, computed as $T_{RC} T_{CO}^0$, where $T_{CO}^0$ ​is the object pose relative to the camera. This pose is estimated by our method and corresponds to the first frame of the video. $T_{RC}$ is the virtual camera pose relative to the robot, which was manually chosen to simulate a camera looking at the robot from the front with a 30-degree elevation relative to the gravity axis. This camera pose was manually defined and kept constant across all videos. Using this approach, the object was moved to a pose visually similar to that shown in the videos. However, in practice, this step would not be necessary as the robot would start with the object already grasped in its gripper or a separate grasping process would be called (e.g. using a combination of motion planning and GraspIT/GraspNet)*.*
>
> **Following the trajectory from the video:** In the second step, we computed the object's motion. To increase the transferability of the method, we first expressed the object's motion relative to the starting pose of the extracted object trajectory from the video. This relative motion was then applied to the object's pose in simulation (or real life) to derive the reference object trajectory for the robot. Finally, trajectory optimization, as shown in Eq. (4), was solved to obtain the motor torques of the robot to imitate the reference object trajectory
>
> **Quantitative evaluation:** Quantitative evaluation is indeed missing, as measuring success rates for in-the-wild object interactions is challenging. In our real-robot experiments, all processed videos were successfully retargeted to the robot, and all object motions resulted in putting some material inside the (manually placed) target object. While this could serve as a measure of success, we felt it was too coarse to report it. Therefore, we limited ourselves to qualitative evaluation for the robotic experiments. However, we would be happy to report it in the revised version of the paper.
>
> > Fig. 8, in the upper additional qualitative result, the bowl object pose is not correctly tracked. Why does the robot still turn the object in a quite different angle?
>
> **Response:**
> Thank you for letting us clarify this issue. Please note that the input video, robot simulation, and real-robot videos were captured from different camera viewpoints. The same robot trajectory is executed in the simulation and by the real robot.
>
> The robot trajectory is always computed to closely imitate the relative 6D pose transformations of the detected object from the video. We only transform the trajectory from the camera frame into the robot frame to account for differences in scale and robot position as described in response to point above.
> Because of this, the perceived differences between the estimated object trajectory from the video, the object trajectory executed by the robot in the simulation and the object trajectory executed by the real robot are due to the differences of the viewpoints.. We hope this clarifies the issue.
>
> ### **Additional minor comments**
>
> > Fig. 6, rightmost real robot image seems to be a repetition of the image next to it.  Was the wrong image included?
>
> **Response:**
> Thank you for noticing, this was indeed a mistake. We will replace the image with the correct one in the revised version.
>
> ### **Questions**
>
> > l. 323, are the ground-truth meshes contained in the object datasets?
>
> **Response:**
> The exact ground-truth meshes are most likely not part of Objaverse (at least we did not see them). However, the dataset can either contain other 3D meshes of the same objects, or meshes that are very similar (e.g. Campbell's Primordial Soup instead of the Tomato Soup \- see Figure 3).
>
> > Table 1, was the same scale estimate for the meshes used for MegaPose and GigaPose like for the proposed method?
>
> **Response:**
> Yes, to consistently compare all 6D pose estimation methods mentioned in Table 1, we always reuse the same object detections and respective scale estimates.

---

> ### Author Response · Authors · 2024-11-21
>
> > Which dynamics model is used for the optimization problem in eq 4? How is tracking of the optimized trajectory implemented?
>
> **Response:**
> We utilized the publicly available dynamic model for the Panda robot from the example-robot-data package, accessible via Conda and PyPI. This package includes the kinematics, as well as the geometric and inertia parameters for each link of the robot.
>
> For forward dynamics computation, we employed the Pinocchio library, which is also internally used within the Aligator trajectory optimization package. The dynamic model was used to compute joint positions and velocities based on the input sequence of joint torques, while the kinematic model was used to determine the object's pose through forward kinematics. Consequently, both the end-effector pose and joint velocities are directly influenced by the optimized torques in Eq. (4).
>
> The robot is controlled in joint space (with 7 revolute joints) in position mode, defined by joint angles. To ensure smooth motion, joint torques were optimized using Eq. (4), and the corresponding joint positions were computed based on the forward dynamics.

---

> ### Comment · Reviewer_JJFi · 2024-11-25
> **Thanks for author response**
>
> The author response addressed most of my concerns well.
>
> Wrt. quantitative evaluation: the result reported in the author response indeed seems too coarse for a scientific paper. Instead, a quantifiable measure of success and a repeatable set of test scenarios should be defined and evaluated.

---

> > ### Author Response · Authors · 2024-11-30
> >
> > Thank you for your suggestion. We completely agree that establishing quantifiable measures of success and a repeatable experimental setup is crucial for making progress on this challenging problem. To address this, we have developed an annotation tool designed to annotate object manipulation in internet videos, providing approximate trajectories for the manipulated objects.
> >
> > **Annotation.** Specifically, our approach involves precomputing the top 25 candidates for mesh retrieval, after which the annotator manually selects the best-fitting mesh for the given video. Symmetry is manually annotated to indicate if the selected object is symmetric (e.g., a bowl), in a similar manner as done in Labbe et al., ECCV 2020 ([https://arxiv.org/abs/2008.08465](https://arxiv.org/abs/2008.08465)). The selected mesh is then manually aligned with the manipulated object in each video frame. During this process, the annotator adjusts the x-y translation, depth, SO(3) rotations, and perspective effects while assuming an approximate scale for the mesh (set at 15 cm for the annotated objects). This methodology ensures an approximate reconstruction of the full SE(3) trajectory for the annotated object, providing a solid foundation for further evaluation. So far, we have used this annotation tool to compute reference trajectories for five Internet videos (altogether \~900 frames), which are presented qualitatively in the submitted manuscript.
> >
> > **Metric.** For the evaluation, we designed metrics that compare the relative transformations of objects over time, ensuring robustness to variations in the starting poses of the motion and emphasizing the nature of the motion rather than its absolute position. The evaluation is split into two components: rotation and translation.
> >
> > To assess rotation, we compute the spatial angular velocities required to rotate the object from frame \\( k \\) to frame \\( k \+ \\Delta \\). We then compare the angular velocities of the manually annotated reference trajectory with those of the trajectories produced by our method or a baseline. The metric is averaged across all frames \\( k \\) for various values of \\( \\Delta \\), ranging from 1 (to evaluate local consistency) to 50% of the trajectory length (to evaluate global consistency). To deal with symmetries, a minimal error is selected over the annotated symmetries. The spatial velocity is used instead of the body velocity to ensure that the metric is agnostic to the choice of the object’s coordinate frame, maintaining consistency regardless of variations in the object's body frame orientation or location.
> >
> > For the translation evaluation, we assess separatelly (i) the spatial velocities of object positions projected onto the image plane and (ii) the scaled depth velocities. We use the projected spatial velocities to factor out the effect of scale/depth ambiguity. For the depth velocity computation, we normalize depth by the scale of the object using 15 cm for the annotated ground truth and the absolute scale estimated by our method for the retrieved mesh. The same averaging approach used for rotation is applied here, averaging over all frames in the video and across various values of \\( \\Delta \\).

---

> > ### Author Response · Authors · 2024-11-30
> >
> > **Results.** We used the annotated videos and defined metrics to compute the rotation, projected translation, and scaled depth errors for our method and the MegaPose baseline, as shown in the table below. This initial small-scale evaluation demonstrates that our method outperforms the MegaPose baseline in all dimensions. Additionally, our pose-tracking approach improves performance compared to our per-frame evaluation by introducing temporal filtering through tracklets.
> >
> > It is worth noting that the projected translation errors are identical for MegaPose (coarse) and our method (per-frame), as both approaches use the same technique for translation estimation. However, our pose-tracking approach slightly improves the translation error compared to the MegaPose refiner, which suffers from inaccuracies when its assumption of an identical mesh is violated.
> >
> > These preliminary results from the small-scale evaluation suggest that our approach is superior. However, due to limited time and the time-consuming nature of the annotation process, we were unable to annotate additional videos for the rebuttal. To strengthen this quantitative evaluation of our method, we are currently working on expanding the dataset to include 20 Internet videos of humans performing actions with everyday objects. This will enable us to compare our method more robustly. We will also compare against other baselines used in the submitted manuscript. Overall, we plan to annotate ~100 Internet videos to form an open-source test set to enable measuring progress on this hard problem.
> >
> > |  | MegaPose Coarse (per-frame) | MegaPose Coarse+Refine (per-frame) | Ours  (per frame) | Ours  (pose tracking) |
> > | :---- | :---- | :---- | :---- | :---- |
> > | Average relative rotation \[deg\] | 59.61 | 58.14 | 19.17 | 16.03 |
> > | Average relative projected translation \[px\] | 28.08 | 45.07 | 28.08 | 26.53 |
> > | Average relative scaled depth | 1.38 | 1.39 | 1.38 | 0.98 |

---

### Official Review · Reviewer_x6Vw · 2024-11-03

**Soundness:** 3
**Presentation:** 3
**Contribution:** 3
**Rating:** 6
**Confidence:** 4

**Summary:**

The paper introduces a pipeline for extracting the 6D pose trajectory from an internet video without the need of the CAD for the specific object. The authors leverage vision features to retrieve the most similar CAD model of the object, then do per-frame alignment leveraging the same vision features of the original image and rendered from the CAD. They further estimate the rough object size using LLM and leverage 2D tracking models to get inter-frame rotation consistency. The authors conduct experiments and demonstrate their superior performance. They also show demos that their trajectory can be retargeted to guide the movement of the robot.

**Strengths:**

1. The task of predicting the 6D pose of internet videos without additional prior is important for a lot of downstream tasks.
2. The whole pipeline is reasonable, fetch the similar CAD model and do rough alignment. Then further leverage the 2D tracking results to get the smoothed trajectories, that are more motion-consistent across time.
3. The experiments on the retargeted motion on robotics further show the usefulness of the extracted smoothed trajectories.

**Weaknesses:**

1. The authors demonstrate that compared to model-based methods, whose performances suffer from the inaccurate CAD mode, their method addresses the challenge. However, there is lack of experiments compared to SOTA model-based methods with their fetched CAD models (e.g. FoundationPose with their retrieved CAD model).
2. In the 6D pose alignment part, the method applies a sapling-based trajectory to get the rotation, which potentially limits the accuracy of the rotation. In the results figure, there are some rotation errors, not sure if due to the sampling-based strategy or the DINO feature extractor.
3. For the robotics demo, the end-effector position control is on 6D pose or only on the rotation? From the Figure 9, the translation of the end-effector seems not consistent with the original video and in the simulator

**Questions:**

1. Why in Figure 1 and Figure 2, the same image has two different retrieved CAD models?
2. Can you provide the results of the error based on the quality of the retrieved CAD model?

---

> ### Author Response · Authors · 2024-11-21
>
> ### **Weaknesses**
> > The authors demonstrate that compared to model-based methods, whose performances suffer from the inaccurate CAD mode, their method addresses the challenge. However, there is lack of experiments compared to SOTA model-based methods with their fetched CAD models (e.g. FoundationPose with their retrieved CAD model).
>
> **Response:**
> We compared our approach against state-of-the-art methods from the BOP challenge in the “unseen-objects” category. Our target application is YouTube videos, where depth measurements are not accessible, so we limit our comparison to RGB-only inputs. The top methods from the BOP challenge are Co-op (GenFlow based) \+ GenFlow \[A\], GigaPose \[B\], FoundPose \[C\], and MegaPose \[D\]. Please note that FoundationPose is not part of the list as it is RGBD-based.
>
> Unfortunately, the open-source implementation of Co-op/GenFlow is not available. In the paper, we show a comparison with GigaPose and MegaPose (see Tab. 1). At the time of submission, the open-source implementation of FoundPose was not available, so we verified one of its key ideas (the Bag-of-Words approach) on our method and observed that it did not perform well with inaccurate CAD models. The FoundPose code is now available, and we evaluate it in the table below. It can be seen that while the implementation, which includes their translation estimation, improves the results, FoundPose still suffers from inaccuracies in the mesh compared to our proposed approach. We would be happy to compare with any other publicly available approach.
>
> \[A\] Moon, Sungphill, et al. "Genflow: Generalizable recurrent flow for 6d pose refinement of novel objects." *Proceedings of the IEEE/CVF Conference on Computer Vision and Pattern Recognition*. 2024\.
> \[B\] Nguyen, Van Nguyen, et al. "Gigapose: Fast and robust novel object pose estimation via one correspondence." *Proceedings of the IEEE/CVF Conference on Computer Vision and Pattern Recognition*. 2024\.
> \[C\] Örnek, Evin Pınar, et al. "Foundpose: Unseen object pose estimation with foundation features." *European Conference on Computer Vision*. Springer, Cham, 2025\.
> \[D\] Labbé, Yann, et al. "MegaPose: 6D Pose Estimation of Novel Objects via Render & Compare." *CoRL 2022-Conference on Robot Learning*. 2022\.
>
> |  | YCB-Video |  |  |  | HOPE-Video |  |  |  |
> | :---- | ----- | :---- | :---- | :---- | ----- | :---- | :---- | :---- |
> | Method | AR | AR$_{CoU}$ | AR$_{CH}$ | AR$_{pCH}$ | AR | AR$_{CoU}$ | AR$_{CH}$ | AR$_{pCH}$ |
> | MegaPose (w/o refiner) | 23.75 | 10.08 | 10.65 | 50.53 | 31.77 | 9.96 | 6.87 | 78.50 |
> | MegaPose | 25.76 | 14.01 | 11.91 | 51.37 | 33.03 | 13.07 | 6.38 | 79.64 |
> | GigaPose | 29.18 | 11.90 | 9.20 | 66.45 | 23.12 | 4.15 | 4.90 | 60.30 |
> | FoundPose (full) | 42.95 | 35.40 | 15.69 | 77.75 | 42.30 | 31.18 | 9.58 | 86.13 |
> | Ours | **49.86** | **45.20** | **18.53** | **85.83** | **45.98** | **39.21** | **10.72** | **88.01** |
>
> > In the 6D pose alignment part, the method applies a sapling-based trajectory to get the rotation, which potentially limits the accuracy of the rotation. In the results figure, there are some rotation errors, not sure if due to the sampling-based strategy or the DINO feature extractor.
>
> **Response:**
> To examine whether rotation errors are caused by sampling, we perform a new ablation study on YCB-V dataset in which we increase the number of sampled views in the utilized sampling strategy \[SuperFib\]. In our method, we use N \= 600 samples, which, on average, lead to a \~25-degree geodesic error between the closest rotations. When increasing N to 1200 or 1800, we observe that the overall pipeline performance remains similar (within a statistical error), while the computational requirements for storage and runtime increase linearly. Our prerendered mesh database takes \~1TB of disk space for N=600 and \~2 TB for N=1200 views and \~3 TB for N \= 1800 views. The runtime also scales linearly from \~0.2s per object to \~0.4s per object for 1200 views and \~0.6s per object for 1800 views.
>
> | N$_{samples}$ | AR | AR$_{CoU}$ | AR$_{CH}$ | AR$_{pCH}$ | Avg Err |
> | :---- | :---- | :---- | :---- | :---- | :---- |
> | 600 | 49.86 | 45.20 | 18.53 | 85.83 | \~25 deg |
> | 1200 | 49.66 | 44.64 | 19.10 | 85.24 | \~20 deg |
> | 1800 | 49.10 | 43.95 | 18.25 | 85.11 | \~16 deg |
>
> \[SuperFib\] Marc Alexa, Super-Fibonacci Spirals: Fast, Low-Discrepancy Sampling of SO(3), CVPR2022

---

> ### Author Response · Authors · 2024-11-21
>
> > For the robotics demo, the end-effector position control is on 6D pose or only on the rotation? From the Figure 9, the translation of the end-effector seems not consistent with the original video and in the simulator
>
> **Response:**
>
> Thank you for letting us clarify this issue. Please note that the input video, robot simulation, and real-robot videos were captured from different camera viewpoints. The same robot trajectory is executed in the simulation and by the real robot.
>
> The robot trajectory is always computed to closely imitate the full 6D pose of the detected object from the video. We only manually transform the trajectory from the camera frame into the robot frame to account for differences in scale and robot position.
> Because of this, the perceived differences between the estimated object trajectory from the video, the object trajectory executed by the robot in the simulation and the object trajectory executed by the real robot are due to the differences of the viewpoints used for the visualization. We hope this clarifies the issue.
>
>
> ### **Questions**
>
> > Why in Figure 1 and Figure 2, the same image has two different retrieved CAD models?
>
> **Response:**
> We apologize for any confusion caused. Our intention with the overview figure (Fig. 2\) was solely to illustrate the entire pipeline. As such, we have used one of the visually pleasing meshes among the top retrieved meshes instead of using the top retrieval from the Objaverse database. In the revised version of the paper, we will incorporate the retrieved mesh to prevent any potential future misunderstandings.
>
> > Can you provide the results of the error based on the quality of the retrieved CAD model?
>
> **Response:**
>
> In general, it is difficult to define the quality of the retrieved CAD model. As a proxy, we compare the quality of the recovered pose (i) using the retrieved mesh and  (ii) using the ground truth exact mesh (which is known for the standard datasets). This allows us to approximately analyze the metric sensitivity to the quality of the CAD model. Results are shown in Table 2 in the main paper, which is also shown below for your convenience. The last row of the table presents the average recall for the Oracle across various metrics (columns). The Oracle represents a scenario where both the mesh model and its corresponding scale are known from ground truth. The other rows present different methods for mesh retrieval.
>
> The CoU (complement over union) and pCH (projected chamfer distance) metrics measure projected information, which is not significantly affected by depth. The Oracle mesh performs the best, with a reasonable decrease in performance for the other retrieval methods. This roughly illustrates the sensitivity of the metrics to the quality of the mesh.
>
> The remaining metric, CH (Chamfer distance), measures the distance between vertices in 3D space. A significant drop in performance is observed for all methods compared to the Oracle. This drop is primarily caused by the scale estimation method, which affects the distance between the object and the camera.
>
> | Retrieval | AR | AR$_{CoU}$ | AR$_{CH}$ | AR$_{pCH}$ |
> | :---- | :---- | :---- | :---- | :---- |
> | (a) OpenShape | 34.51 | 20.25 | 10.59 | 72.69 |
> | (b) Ours (CLS) | 45.05 | 40.93 | 16.16 | 78.06 |
> | (c) Ours | 49.86 | 45.20 | 18.53 | 85.83 |
> | (d) Oracle | 62.93 | 51.99 | 45.95 | 90.85 |

---

> > ### Comment · Reviewer_x6Vw · 2024-11-24
> >
> > Thanks for the response from the authors. Most of my concerns are addressed. I will raise my score.

---

### Meta-Review · Area_Chair_yotE · 2024-12-17

**Metareview:**

The paper introduces a novel framework for extracting temporally consistent 6D pose trajectories of manipulated objects from internet videos without requiring CAD models. The authors address the challenges by combining CAD retrieval, pose estimation, and trajectory optimization techniques. Experiments on video datasets demonstrate state-of-the-art performance. The real-world robotic manipulation demo further validates the practical utility of the approach. Reviewers consistently agree with the novelty of the pipeline, its practical significance, and its robust evaluation. During the discussion, concerns from the reviewers have been resolved with further details.

**Additional Comments On Reviewer Discussion:**

There were several rounds of discussions, and most of the concerns from the reviewers were addressed. For example, 1) the authors clarified the comparison with state-of-the-art models, highlighted the advantages of their approach; 2) the authors demonstrated the effective use of a CAD model. Additionally, the authors responded to various questions raised by reviewers to improve the paper's quality in the revision.

---

### Decision · Program_Chairs · 2025-01-22

Accept (Poster)